# HyperSINDy: Deep Generative Modeling of Nonlinear Stochastic Governing Equations

## Abstract

The discovery of governing differential equations from data is an open frontier in machine learning. The *sparse identification of nonlinear dynamics* (SINDy) (Brunton et al., 2016) framework enables data-driven discovery of interpretable models in the form of sparse, deterministic governing laws. Recent works have sought to adapt this approach to the stochastic setting, though these adaptations are severely hampered by the curse of dimensionality. On the other hand, Bayesian-inspired deep learning methods have achieved widespread success in high-dimensional probabilistic modeling via computationally efficient approximate inference techniques, suggesting the use of these techniques for efficient stochastic equation discovery. Here, we introduce *HyperSINDy*, a framework for modeling stochastic dynamics via a deep generative model of sparse governing equations whose parametric form is discovered from data. HyperSINDy employs a variational encoder to approximate the distribution of observed states and derivatives. A hypernetwork (Ha et al., 2016) transforms samples from this distribution into the coefficients of a differential equation whose sparse form is learned simultaneously using a trainable binary mask (Louizos et al., 2018). Once trained, HyperSINDy generates stochastic dynamics via a differential equation whose coefficients are driven by a Gaussian white noise. In experiments, HyperSINDy accurately recovers ground truth stochastic governing equations, with learned stochasticity scaling to match that of the data. Finally, HyperSINDy provides uncertainty quantification that scales to high-dimensional systems. Taken together, HyperSINDy offers a promising framework for model discovery and uncertainty quantification in real-world systems, integrating sparse equation discovery methods with advances in statistical machine learning and deep generative modeling.

## 1 Introduction

Across numerous disciplines, large amounts of measurement data have been collected from dynamical phenomena lacking comprehensive mathematical descriptions. It is desirable to model these data in terms of governing equations involving the state variables, which typically enables insight into the physical interactions in the system. To this end, recent years have seen considerable progress in the ability to distill such governing equations from data alone (e.g., (Schmidt & Lipson, 2009; Brunton et al., 2016)). Nonetheless, this remains an outstanding challenge for systems exhibiting apparently stochastic nonlinear behavior, particularly when lacking even partial knowledge of the governing equations. Such systems thus motivate probabilistic approaches that not only reproduce the observed stochastic behavior (e.g., via generic stochastic differential equations (SDEs) (Friedrich et al., 2011) or neural networks (Girin et al., 2021; Lim & Zohren, 2021)), but do so via discovered analytical representations that are parsimonious and physically informative (Boninsegna et al., 2018).

We are particularly interested in model-free methods that seek to discover both the parameters and functional form of governing equations describing the data. To this end, the sparse identification of nonlinear dynamics (SINDy) framework (Brunton et al., 2016) has emerged as a powerful data-driven approach that identifies both the coefficients and terms of differential equations, given a pre-defined library of candidate functions. The effectiveness of SINDy for sparse model discovery derives from the tendency of physical systems to possess a relatively limited set of active terms. Extensions of the SINDy framework have sought to increase its robustness to noise, offer uncertainty quantification

(UQ), and make it suitable for modeling stochastic dynamics (Boninsegna et al., 2018; Niven et al., 2020; Messenger & Bortz, 2021; Hirsh et al., 2021; Callaham et al., 2021; Fasel et al., 2022; Wang et al., 2022). However, these extensions have generally relied upon computationally expensive approaches to learn the appropriate probability distributions. As such, a unified and computationally tractable formulation of SINDy that meets these additional goals is presently lacking.

Variational inference (VI) methods represent a class of techniques for addressing the complex and often intractable integrals arising in exact Bayesian inference, instead approximating the true posterior via simple distribution(s). Recently, the combination of *amortized* VI (Ganguly et al., 2022) with the representational capacity of neural networks has emerged as a powerful, efficient approach to probabilistic modeling, with widespread application in the form of deep generative models (Kingma & Welling, 2014; Rezende & Mohamed, 2015). Despite the success of these approaches for dynamical modeling (e.g., (Girin et al., 2021)), applications thus far have utilized generic state space formulations or parameter inference on a known functional form of the dynamics. Thus, the potential for VI to facilitate probabilistic equation discovery remains largely unexplored.

## 1.1 CONTRIBUTIONS

In this work, we propose HyperSINDy, a VI-based SINDy implementation that learns a parameterized distribution of ordinary differential equations (ODEs) sharing a common sparse form. Specifically, HyperSINDy employs a variational encoder to parameterize a latent distribution over observed states and derivatives, then uses a hypernetwork (Ha et al., 2016; Pawlowski et al., 2018) to translate samples from this distribution into the coefficients of a sparse ODE whose functional form is learned in a common optimization. In this way, HyperSINDy is able to model complex stochastic dynamics through an interpretable analytical expression – technically, a *random* ODE (Han & Kloeden, 2017) – whose coefficients are parameterized by a white noise process.

Specific contributions of the HyperSINDy framework include:

- **Efficient and Accurate Modeling of Stochastic Dynamics at Scale.** Through VI, we circumvent the curse of dimensionality that hampers other methods in identifying sparse stochastic equations. Specifically, HyperSINDy can accurately discover governing equations for stochastic systems having well beyond two spatial dimensions, which existing approaches have not exceeded (e.g., (Boninsegna et al., 2018; Callaham et al., 2021; Wang et al., 2022; Huang et al., 2022)).

- **Generative Modeling of Dynamics.** Once trained, HyperSINDy generates a random dynamical system whose vector field is parameterized by a Gaussian white noise. Hence, our approach efficiently arrives at a generative model for both the system dynamics and the exogenous disturbances (representing, e.g., unresolved scales). This permits simulations that reproduce the stochastic dynamical behavior of the observed process, while providing a natural method for quantifying uncertainty of the model parameters and propagating uncertainty in the probabilistic model forecast.

- **Interpretable Governing Equations Discovery.** In contrast to other deep generative approaches for modeling stochastic dynamics, HyperSINDy discovers the analytical form of a sparse governing equation without a priori knowledge. Sparsity promotes human readable models where each term corresponds to an interpretable physical mechanism. This notion of interpretability, based on sparsity, is appealing in the traditional perspective of engineering and physics.

In section 1.2, we discuss relevant literature. In section 2, we provide a background on the specific methods and mathematics that inspired our method. In section 3, we describe HyperSINDy. In section 4, we show results on various experiments. In section 5, we conclude with a discussion of our method, its limitations, and possible future directions.

## 1.2 RELATED WORK

HyperSINDy bridges two parallel lines of work concerning data-driven modeling for stochastic dynamics: namely, probabilistic sparse equation discovery and deep generative modeling.

Most probabilistic implementations of SINDy have concerned UQ and noise robustness in the deterministic setting, rather than modeling stochastic dynamics per se. Of these approaches, ensembling methods (E-SINDy) (Fasel et al., 2022) have achieved state-of-the-art UQ and noise robustness for deterministic SINDy models, and were recently shown (Gao et al., 2023) to offer a computationally

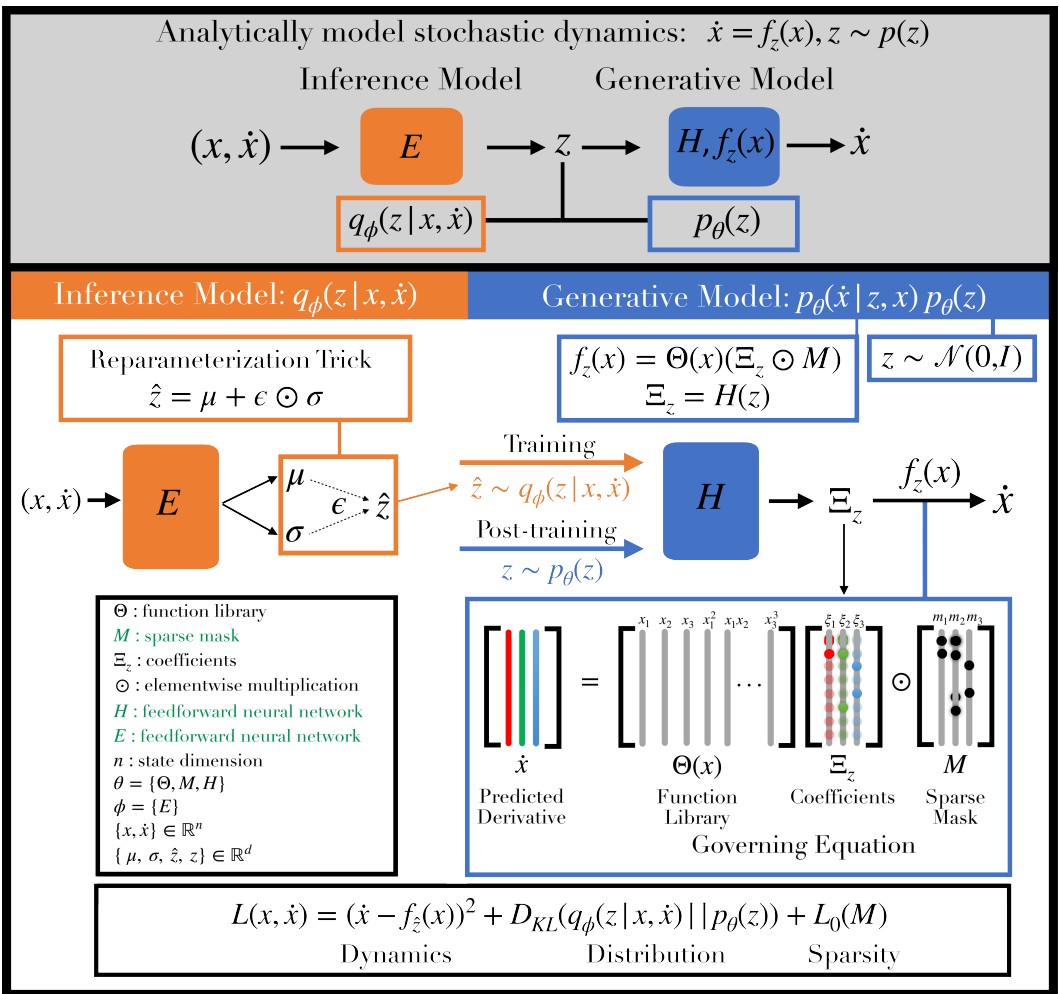

Figure 1: **HyperSINDy Framework**. HyperSINDy employs an inference model and generative model to discover an analytical representation of observed stochastic dynamics in the form of a random (nonlinear) ODE $f_{\mathbf{z}}(x)$. The inference model is an encoder neural network that maps $(\mathbf{x}, \dot{\mathbf{x}})$ to the parameters $\mu$ and $\sigma$ of $q_\phi(\mathbf{z}|\mathbf{x}, \dot{\mathbf{x}})$. $\hat{\mathbf{z}}$ can be sampled using a simple reparameterization of $\mu$ and $\sigma$. The generative model predicts the derivative via a hypernetwork $H$, which transforms $\mathbf{z}$ into $\Xi_{\mathbf{z}}$, the coefficients of the ODE. $f_{\mathbf{z}}(\mathbf{x})$ comprises a function library $\Theta$, the coefficients $\Xi_{\mathbf{z}}$, and sparse mask $M$. Once trained, stochastic dynamics can be generated by iteratively sampling $\mathbf{z}$ from its prior $\mathbf{z} \sim p_\theta(\mathbf{z})$, thus yielding new sample paths of the noise-paramterized vector field, $\Xi_{\mathbf{z}}$. In the legend, trainable parameters are shown in green. The loss function comprises terms related to 1) the derivative reconstructions, 2) the latent distribution $q_\phi(\mathbf{z}|\mathbf{x}, \dot{\mathbf{x}})$, and 3) sparsity of the discovered equation. See accompanying pseudocode 1 and 2 for details on batch-wise training.

efficient alternative to earlier Bayesian implementations of SINDy (Niven et al., 2020; Hirsh et al., 2021) leveraging costly sampling routines to compute posterior distributions. Nonetheless, a model of the process noise is crucial for accurate UQ in the stochastic dynamics setting. Multiple studies have generalized the SINDy framework for the identification of parametric SDEs (Boninsegna et al., 2018; Callaham et al., 2021), with three such studies recently performed in the Bayesian setting (Wang et al., 2022; Huang et al., 2022; Tripura & Chakraborty, 2023). However, as discussed in these works, existing methods for approximating the drift and diffusion terms of the SDE (e.g., constructing histograms for the Kramers-Moyal expansion) are severely hampered by the curse of dimensionality, with computational cost generally scaling exponentially with SDE state dimension. Thus, an efficient and scalable formulation of SINDy for stochastic dynamics remains lacking.

A separate line of work has leveraged advances in probabilistic deep learning for modeling stochastic dynamics, with deep generative models achieving state-of-the-art performance across a range of modeling tasks ((Yoon et al., 2019; Girin et al., 2021)). Although these models do not typically involve explicit dynamical representations, advances in physics-informed machine learning (Karniadakis et al., 2021) have motivated numerous developments at this intersection (e.g., (Lopez & Atzberger, 2021; Takeishi & Kalousis, 2021; Yang et al., 2020; Zhang et al., 2019). As for (stochastic) equation discovery, several recent works have successfully employed VAEs to learn the coefficients of a generic (or pre-specified) SDE within a (potentially lower-dimensional) latent space(Hasan et al., 2022; García et al., 2022; Nguyen et al., 2021; Zhong & Meidani, 2023). We propose to similarly leverage a VAE-like architecture to perform inference on a latent stochastic process; however, we seek to additionally discover a structural representation of the governing laws, which can yield considerable physical insight into the system (Boninsegna et al., 2018; Nayek et al., 2021; Wang et al., 2022). In sum, we seek to bridge the above fields via a unified deep learning architecture (trainable end-to-end with backpropagation) that enables discovery of the functional form of a governing stochastic process, along with posterior distributions over the discovered system coefficients (e.g., for UQ).

## 2 Background

We briefly overview the SINDy and VAE frameworks, as well as an implementation of an $L_0$ loss, before describing their integration within the HyperSINDy architecture.

**Sparse Identification of Nonlinear Dynamics**   The SINDy (Brunton et al., 2016) framework leverages sparse regression to enable discovery of a parsimonious system of differential equations from time-ordered snapshots. Thus, consider a system with state $\mathbf{x}(t) \in \mathbb{R}^d$ governed by the ODE:

$$\dot{\mathbf{x}}(t) = f(\mathbf{x}(t)) \tag{1}$$

Given $m$ observations of the system in time $\mathbf{X} = [\mathbf{x}(t_1), \mathbf{x}(t_2), ..., \mathbf{x}(t_m)]^T$ and the estimated time derivatives $\dot{\mathbf{X}} = [\dot{\mathbf{x}}(t_1), \dot{\mathbf{x}}(t_2), ..., \dot{\mathbf{x}}(t_m)]^T$, we construct a library of candidate functions $\Theta(\mathbf{X}) = [\theta_1(\mathbf{X}), \theta_2(\mathbf{X}), ..., \theta_l(\mathbf{X})]$. We then solve the regression problem, $\dot{\mathbf{X}} = \Theta(\mathbf{X})\mathbf{\Xi}$, to identify the optimal functions and coefficients in $\Theta$ and $\mathbf{\Xi}$, respectively. A sparsity-promoting regularization function $R$ is typically added to this model discovery problem, yielding the final optimization, $\hat{\mathbf{\Xi}} = \arg\min_{\mathbf{\Xi}} (\dot{\mathbf{X}} - \Theta(\mathbf{X})\mathbf{\Xi})^2 + R(\mathbf{\Xi})$. Although we focus on this basic implementation, we note that there have been numerous extensions of the original SINDy framework (for a recent overview, see (Kaptanoglu et al., 2022)), many of which can be easily incorporated into the present framework.

**Variational Autoencoder**   The VAE framework (Kingma & Welling, 2014) elegantly integrates variational inference (VI) with deep learning architectures, providing an efficient and powerful approach toward probabilistic modeling. VAEs assume that a set of observations $\mathbf{x}$ derives from a corresponding set of latent states $\mathbf{z}$. VAEs construct an approximate posterior distribution $q_\phi(\mathbf{z}|\mathbf{x})$ and maximize the evidence lower bound (ELBO) of the log likelihood of the data $p_\theta(\mathbf{x})$:

$$\log p_\theta(\mathbf{x}) \geq ELBO(\mathbf{x}, \mathbf{z}) = \mathbb{E}_{q_\phi(\mathbf{z}|\mathbf{x})}[\log p_\theta(\mathbf{x}|\mathbf{z})] - D_{KL}(q_\phi(\mathbf{z}|\mathbf{x})||p_\theta(\mathbf{z})) \tag{2}$$

where $\phi$ and $\theta$ and are the parameters of the inference (encoder) and generative (decoder) models, respectively. The "reparameterization trick" enables sampling from $q_\phi(\mathbf{z}|\mathbf{x})$ using $\mathbf{z} = \mu(\mathbf{z}) + \sigma(\mathbf{z}) \odot \epsilon$ while still training the network end-to-end with backpropagation. After training, new observations are easily generated by sampling from the prior $p_\theta(\mathbf{z})$, typically a unit Gaussian with diagonal covariance.

$L_0$ **Regularization**   The $L_0$ norm is ideal for sparse regression problems as it penalizes all nonzero weights equally, regardless of magnitude. As $L_0$ regularization poses an intractable optimization problem, the $L_1$ regularization (lasso) – which penalizes the actual values of the learned weights – is a more common technique to achieve sparsity in practice. Nonetheless, incorporation of an $L_0$-norm penalty (Zheng et al., 2019) into SINDy was recently found to have considerable advantages (Champion et al., 2020), motivating us to adopt a backpropagation-compatible $L_0$ regularization. Accordingly, we implement one such method recently proposed by Louizos et al. (2018), which penalizes a trainable mask using the hard-concrete distribution.

Specifically, let $M \in \mathbb{R}^d$ be the desired sparse mask. Let $s$ be a binary concrete random variable (Maddison et al., 2017; Jang et al., 2017) distributed in $(0, 1)$ with probability density $q_\phi(s)$, cumulative density $Q_\phi(s)$, location $\log \alpha$, and temperature $\beta$. Let $\phi = (\log \alpha, \beta)$. Suppose we have $\gamma < 0$

and $\zeta > 1$. We define each element $m$ in $M$ as a hard concrete random variable computed entirely as a transformation of $s$. Thus, learning an optimal $m$ necessitates learning $q_\phi(s)$, which simplifies to optimizing $\log \alpha$ (we fix $\beta$). Sampling from $q_\phi(s)$ and backpropagating into $\log \alpha$ motivates use of the reparameterization trick (as in the VAE above) with $\epsilon \sim \mathcal{U}(0, 1)$. Then, $m$ is computed.

$$s = \text{Sigmoid}((\log \epsilon - \log(1 - \epsilon) + \log \alpha)/\beta) \qquad m = \min(1, \max(0, s(\zeta - \gamma) + \gamma)) \quad (3)$$

After training, we obtain $m$ using our optimized $\log \alpha$ parameter:

$$m = \min(1, \max(0, \text{Sigmoid}(\log \alpha)(\zeta - \gamma) + \gamma)) \tag{4}$$

We train $M$ using the following loss:

$$L_0(M) = \sum_{j=1}^{d} \text{Sigmoid}(\log \alpha_j - \beta \log \frac{\gamma}{\zeta}) \tag{5}$$

Refer to (Louizos et al., 2018) for the full derivation. In short, this provides a backpropagation-compatible approach to enforce sparsity via a trainable, element-wise mask.

## 3 HYPERSINDY

We combine advances in Bayesian deep learning with the SINDy framework to propose HyperSINDy, a hypernetwork (Ha et al., 2016; Pawlowski et al., 2018) approach to parsimoniously model stochastic nonlinear dynamics via a noise-parameterized vector field whose sparse, time-invariant functional form is discovered from data. In brief, HyperSINDy uses a variational encoder to learn a latent distribution over the states and derivatives of a system, whose posterior is regularized to match a Gaussian prior. Once trained, a white noise process generates a time-varying vector field by updating the coefficients of the discovered (random) ODE. Across a range of experiments, new noise realizations generate stochastic nonlinear dynamics that recapitulate the behavior of the original system, while also enabling UQ on the learned coefficients. Fig. 1 provides an overview of our approach and problem setting, which we detail below.

**Problem Setting** Stochastic equations are fundamental tools for mathematically modeling dynamics under uncertainty. In general, the precise physical source of uncertainty is unknown and/or of secondary importance (Friedrich et al., 2011; Duan, 2015; Särkkä & Solin, 2019); as such, several formulations exist. A common choice is the Langevin-type SDE with explicitly separated deterministic (drift) and stochastic (diffusion) terms. Alternatively, we may consider a deterministic ODE with stochastic parameters, i.e., a *random* ODE (RDE), which is another well-established framework (Arnold, 1998; Duan, 2015) with wide-ranging real-world applications (e.g., fluctuating resources in biological systems (Kloeden & Pötzsche, 2013)). Here, we adopt the RDE formulation in the widely studied setting of i.i.d. noise (Arnold, 1998; Caraballo & Han, 2016). We find this formulation practically advantageous for integration with deep generative modeling and VI, enabling a powerful and scalable approach to stochastic dynamics. Importantly, as any (finite-dimensional) SDE can be transformed into an equivalent RDE and vice versa (Han & Kloeden, 2017); these practical advantages can be exploited without compromising relevance to canonical SDE representations (as we will empirically demonstrate).

As above, let $\mathbf{x}_{0:T}$ be the observations from times 0 to $T$ of the state of a system, $\mathbf{x}_t \in \mathbb{R}^n$. We assume these data are generated from some stochastic dynamics $\dot{\mathbf{x}} = f_{\mathbf{z}}(\mathbf{x}_t)$, where $\mathbf{z}$ is a latent random variable modeled as an i.i.d. noise process. We wish to identify a family of sparse vector field functions $f_{\mathbf{z}}$ constrained to a common functional form for all $\mathbf{z} \in \mathbb{R}^d$ (i.e., only the coefficients of $f$ are time-varying, reflecting the system's dependence on fluctuating quantities).

With this framing, we seek to approximate both the functional form $f_{\mathbf{z}}$ and a posterior estimate of the latent noise trajectory $\mathbf{z} = [\mathbf{z}_0, \mathbf{z}_1, ..., \mathbf{z}_T]^T$ associated with each observed trajectory $\mathbf{x}_{0:T}$. To do so, we employ a variational encoder to learn an inference model for the latent space $p(\mathbf{z}|\mathbf{x}, \dot{\mathbf{x}})$ and a generative model $p(\dot{\mathbf{x}}|\mathbf{x}, \mathbf{z})$ subject to $\dot{\mathbf{x}} = f_{\mathbf{z}}(\mathbf{x})$, as detailed below. Ultimately, once trained, we may generate new trajectories of $\mathbf{x}$ simply by iteratively sampling $\mathbf{z}$ from its Gaussian prior (i.e., constructing new sample paths of the driving noise).

**Generative Model**   Consider a factorization of the conditional generative model with parameters $\theta$: $p_\theta(\dot{\mathbf{x}}, \mathbf{z}|\mathbf{x}) = p_\theta(\dot{\mathbf{x}}|\mathbf{z}, \mathbf{x})p_\theta(\mathbf{z})$. We assume that $\mathbf{z}$ is independent of $\mathbf{x}$, so $p_\theta(\mathbf{z}|\mathbf{x}) = p_\theta(\mathbf{z})$. $p_\theta(\dot{\mathbf{x}}|\mathbf{z}, \mathbf{x})$ describes how the state $\mathbf{x}$ and latent $\mathbf{z}$ are transformed into the derivative, while $p_\theta(\mathbf{z})$ is a prior over the latent distribution of states and their derivatives. We take $p_\theta(\mathbf{z})$ to be a standard Gaussian with diagonal covariance: $p_\theta(\mathbf{z}) = \mathcal{N}(0, \mathbf{I})$. We seek to parameterize $p_\theta(\dot{\mathbf{x}}|\mathbf{z}, \mathbf{x})$ according to a nonlinear function, $f_{\mathbf{z}}(\mathbf{x})$. Following the SINDy framework, which seeks interpretability in the form of sparse governing equations, we adapt 1 to obtain the following implementation of $f_{\mathbf{z}}(\mathbf{x})$:

$$f_{\mathbf{z}}(\mathbf{x}) = \Theta(\mathbf{x})(\Xi_{\mathbf{z}} \odot M). \tag{6}$$

where $\odot$ indicates an element-wise multiplication. $\Theta(\mathbf{x})$ is a matrix expansion of $\mathbf{x}$ using a pre-defined library of basis functions, which can include any rational functions, such as polynomial (e.g., $\mathbf{x}_1^2, \mathbf{x}_1\mathbf{x}_2$) or trigonometric (e.g., $\sin \mathbf{x}_1$) functions. $\Xi_{\mathbf{z}}$ is a matrix of coefficients that is output by a hypernetwork $H$ that takes in $\mathbf{z}$ as input: $\Xi_{\mathbf{z}} = H(\mathbf{z})$. $M$ is a matrix of values $M_{ij} \in [0, 1]$ that is trained with a close approximation to a differentiable $L_0$ norm. Specifically, the values of $M$ are simulated using a hard concrete distribution. As such, $M$ enforces sparsity in the terms of each equation through the element-wise multiplication ($\Xi_{\mathbf{z}} \odot M$). See Background for more details.

We constrain $f_{\mathbf{z}}$ to a $d$-parameter family of ODEs sharing a common functional form. Specifically, $H$ may be interpreted as an implicit distribution (Pawlowski et al., 2018) for $p_\theta(\Xi) = \int p_\theta(\Xi|\mathbf{z})p_\theta(\mathbf{z})d\mathbf{z}$. Although we cannot compute the density of $p_\theta(\Xi)$ exactly, we can sample derivative functions by feeding samples $\mathbf{z} \sim p_\theta(\mathbf{z})$ into the hypernetwork: $\Xi_{\mathbf{z}} = H(\mathbf{z})$.

**Inference Model**   Our inference model is defined by the approximate posterior, $q_\phi(\mathbf{z}|\mathbf{x}, \dot{\mathbf{x}})$, with parameters $\phi$. $q_\phi(\mathbf{z}|\mathbf{x}, \dot{\mathbf{x}})$ is implemented by a neural network $E$ and the reparameterization trick, i.e., $\mu_q, \sigma_q = E(\mathbf{x}, \dot{\mathbf{x}}); \hat{\mathbf{z}} = \mu_q + \epsilon \odot \sigma_q$.

**Training**   We train the model end-to-end with backpropagation to minimize the following loss:

$$loss = (\dot{\mathbf{x}} - f_{\hat{\mathbf{z}}}(\mathbf{x}))^2 + \beta D_{KL}(q_\phi(\mathbf{z}|\mathbf{x}, \dot{\mathbf{x}})||p_\theta(\mathbf{z})) + \lambda L_0(M) \tag{7}$$

where $\beta$ and $\lambda$ are hyperparameters. The loss function optimizes the parameters $\phi$ and $\theta$, where $\phi$ are the parameters of $E$ (i.e., the variational parameters) and $\theta$ are the parameters of $H$ and $M$ (note that $p_\theta(\mathbf{z})$ has fixed parameters). Refer to the Appendix for a full derivation of this loss function, and to Background for details on the sparsity-related loss $L_0(M)$ (especially equation 5). To speed up training, every set number of epochs, we permanently set values of $M$ equal to 0 if the magnitude of corresponding coefficients fall below a specific threshold value.

## 4   RESULTS

We evaluate the performance of HyperSINDy on four stochastic dynamical systems. Across a range of (dynamical) noise levels, we seek to assess the accuracy of models identified by HyperSINDy and the degree to which uncertainty estimates faithfully reflect the level of simulated noise. Refer to the Appendix for full details on data generation, training, and simulations.

### 4.1   STOCHASTIC EQUATION DISCOVERY

First, we show results for 3D Stochastic Lorenz and 3D Stochastic Rössler datasets, simulated by:

$$\dot{x} = \omega(y - x) \qquad \dot{y} = x(\rho - z) - y \qquad \dot{z} = xy - \beta z \qquad \text{Lorenz} \tag{8}$$
$$\dot{x} = -y - z \qquad \dot{y} = x + ay \qquad \dot{z} = b + z(x - c) \qquad \text{Rössler} \tag{9}$$

where parameters $(\omega, \rho, \beta)$ and $(a, b, c)$ are each modeled as random processes, simulated by iteratively sampling (at each timestep) from normal distributions with scale $\sigma$ and means $(10, 28, \frac{8}{3})$ and $(0.2, 0.2, 5.7)$, respectively (i.e., each parameter is driven by an independent white noise). We train a HyperSINDy model on three trajectories from each system, with $\sigma = 1, 5, 10$.

We find that HyperSINDy correctly identifies most terms in each equation (Fig. 2). Notably, increasing noise has little impact on the mean coefficients learned by HyperSINDy; instead, the estimated standard deviations of these coefficients proportionately scale with the dynamical noise. Furthermore, HyperSINDy only increases the standard deviation on the terms modeled to have

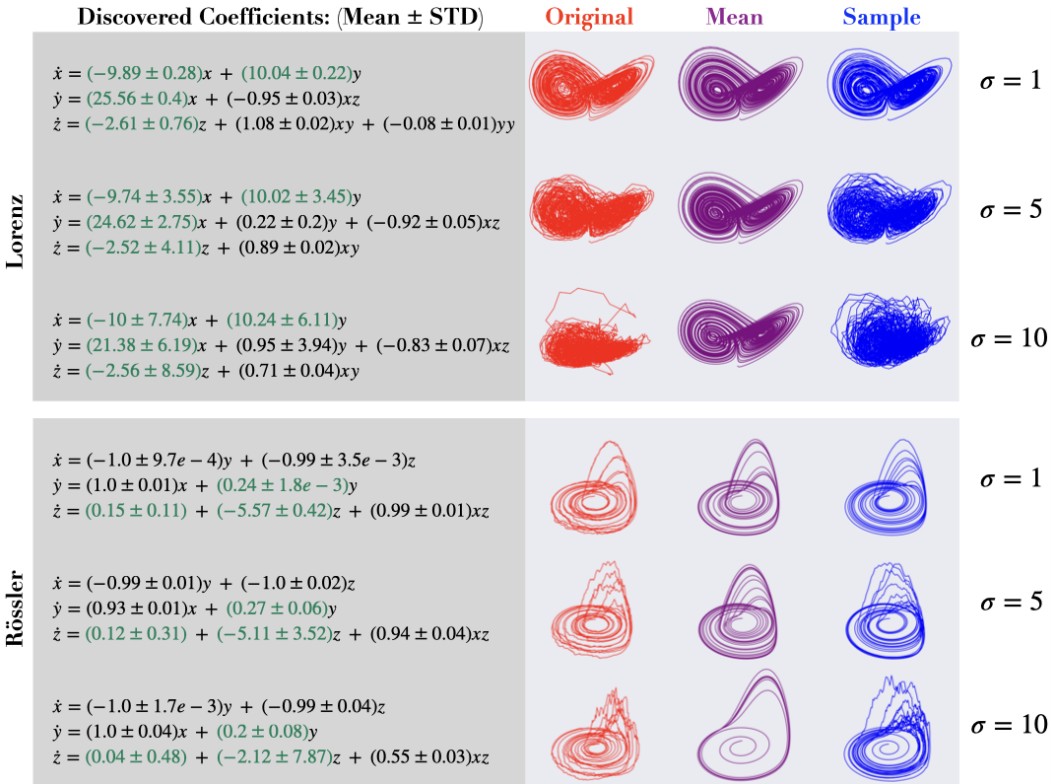

Figure 2: **3D Stochastic Lorenz and Rössler**. HyperSINDy models trained on trajectories simulated with three levels of noise ($\sigma$) for each of the two systems. For each $\sigma$, the mean and standard deviation of the discovered governing equation coefficients are shown (refer to 8 and 9 for ground truth equations). Coefficients in green are those iteratively sampled from Gaussian distributions in the original system. Red trajectories are sample test trajectories simulated with the given $\sigma$. Purple trajectories are generated from HyperSINDy using the mean of the discovered governing equations. Blue trajectories are generated by iteratively sampling from HyperSINDy's learned generative model. See Fig. S1 for additional samples; E-SINDy samples are shown in Fig. S2.

additional noise, while maintaining tight bounds on other terms (e.g. $xy$ in $\dot{y}$ for Lorenz). Moreover, HyperSINDy is able to simulate the original (stochastic) dynamical behavior even as the noise level increases (blue trajectories). On the other hand, because HyperSINDy also successfully identifies the deterministic functional form despite process noise, it is able to produce smooth trajectories (purple) by forecasting with the mean of the discovered equation ensemble.

We sought to benchmark HyperSINDy performance against a leading model for probabilistic model discover, ensemble SINDy (E-SINDy) (Fasel et al., 2022). We trained a total of 30 HyperSINDy and 30 E-SINDy models using Lorenz and Rössler trajectories (see appendix for all simulation details). We evaluated the RMSE of the mean and standard deviation of the discovered equation coefficients, as compared to ground truth. HyperSINDy outperforms E-SINDy on both mean and standard deviation for each experiment (1). Refer to appendix Table S4 for complementary precision and recall results. This pattern of results was minimally sensitive to latent space dimension (Fig. S5). See Table S6 for further quantitative results validating HyperSINDy's strength at equation discovery on a 2D system.

## 4.2 RECOVERING DRIFT-DIFFUSION DYNAMICS

The preceding analyses validate HyperSINDy's capacity for stochastic equation discovery. As HyperSINDy adopts an RDE-based modeling strategy (i.e., a noise-parameterized ODE, as opposed to an SDE with separable drift and diffusion), ground truth equations were explicitly modeled as RDEs rather than SDEs to enable straightforward comparison. As RDEs are conjugate to SDEs

Table 1: Total coefficient RMSE ($\downarrow$) relative to ground truth equations

| Param | | Lorenz | | Rossler | |
|---|---|---|---|---|---|
| | | HyperSINDy | E-SINDy | HyperSINDy | E-SINDy |
| 1 | MEAN | **0.082** $\pm$ 0.004 | 0.18 $\pm$ 0.029 | **0.029** $\pm$ 0.035 | 0.077 $\pm$ 0.04 |
| | STD | **0.598** $\pm$ 0.045 | 1.296 $\pm$ 0.083 | **0.828** $\pm$ 0.059 | 0.849 $\pm$ 0.012 |
| 5 | MEAN | **0.117** $\pm$ 0.022 | 0.268 $\pm$ 0.064 | **0.086** $\pm$ 0.047 | 0.296 $\pm$ 0.199 |
| | STD | **0.4** $\pm$ 0.055 | 0.971 $\pm$ 0.024 | **0.807** $\pm$ 0.012 | 0.875 $\pm$ 0.023 |
| 10 | MEAN | **0.203** $\pm$ 0.047 | 0.349 $\pm$ 0.103 | **0.228** $\pm$ 0.138 | 0.699 $\pm$ 0.551 |
| | STD | **0.279** $\pm$ 0.085 | 0.913 $\pm$ 0.016 | **0.812** $\pm$ 0.014 | 0.875 $\pm$ 0.028 |

Figure 3: **Recovering drift and diffusion behavior in the stochastic Lotka-Volterra model.**. K-M coefficients computed on sample trajectories from each of the three models. From left to right: the ground truth SDE, the HyperSINDy-discovered system, and the Stochastic SINDy-discovered system.

(Han & Kloeden, 2017), this distinction is not fundamental. Nonetheless, this leaves unaddressed the question of how HyperSINDy would learn to represent dynamics explicitly simulated as SDEs.

To address this question, we simulate a 2D SDE to enable direct comparison against the leading method, stochastic SINDy (Boninsegna et al., 2018; Nabeel et al., 2022) (which cannot easily scale to higher dimensions). Specifically, we simulate a widely used model for population dynamics, the stochastic Lotka-Volterra system with state-dependent diffusion:

$$\begin{pmatrix} \mathrm{d}x \\ \mathrm{d}y \end{pmatrix} = \begin{pmatrix} x - xy \\ -y + xy \end{pmatrix} \mathrm{d}t + \begin{pmatrix} \sigma_{xx} & 0 \\ 0 & \sigma_{yy} \end{pmatrix} \begin{pmatrix} \mathrm{d}W_x \\ \mathrm{d}W_y \end{pmatrix} \tag{10}$$

where $\sigma_{xx}(t) = 0.25x - 0.09y$ and $\sigma_{yy}(t) = -0.09x + 0.25y$ give the state-dependent diffusion coefficients, and $W_x(t), W_y(t)$ are independent Wiener processes with i.i.d. increments (i.e., $\Delta W_t = W_{t+1} - W_t \sim \mathcal{N}(0, \Delta t)$). The system is simulated with Euler-Maruyama integration ($\Delta t = 0.01$).

Figure 3 illustrates the results of this analysis. Notably, HyperSINDy learns an expression whose terms correspond to those of the original drift function, thus enabling physical insight into the system. Nonetheless, the analytical expressions are not directly comparable, as the SDE represents stochasticity in terms of a separate diffusion term, while HyperSINDy represents stochasticity as coefficient noise. To enable direct comparison of the deterministic and stochastic aspects of the dynamics discovered by the two methods, we may numerically estimate drift and diffusion coefficients

from the simulated trajectories. Specifically, we may estimate the first two Kramers-Moyal (K-M) coefficients, which derive from a Taylor expansion of the master equation, and which fully describe the Markovian dynamics. Notably, HyperSINDy captures the appropriate deterministic (drift) and stochastic (diffusion) behavior of the system, recapitulating the state-dependence of these terms as seen in the original system – even performing favorably to stochastic SINDy in this setting.

### 4.3 HIGH DIMENSIONAL STOCHASTIC DISCOVERY

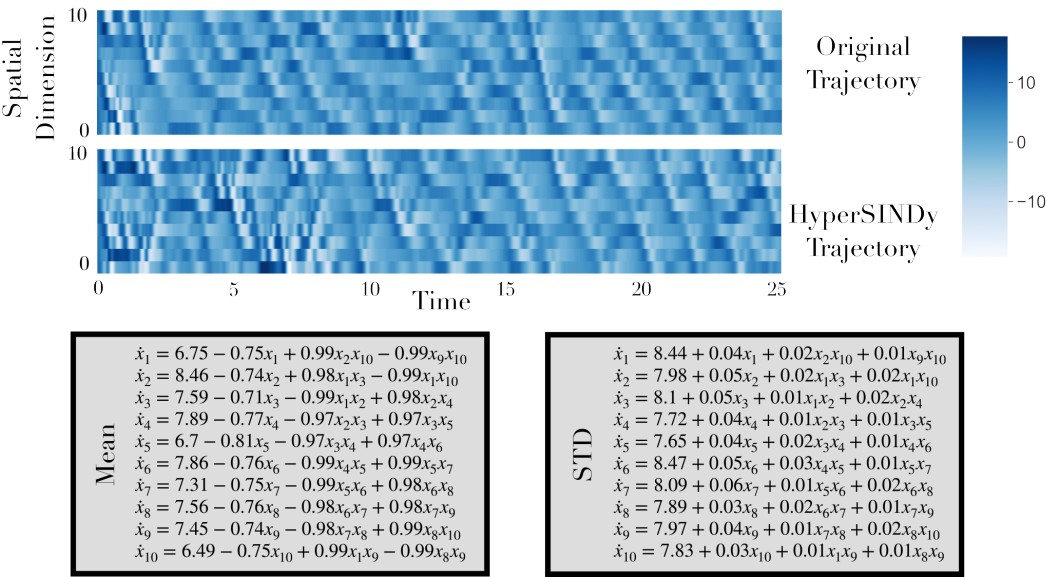

Figure 4: **10D Stochastic Lorenz-96**. A sample test trajectory with $\sigma = 10$ (top) and sample HyperSINDy trajectory (middle) after training on a dataset with $\sigma = 10$. The bottom boxes show the mean and standard deviation of coefficients in the discovered governing equations (cf. Eq. 11).

Lastly, we assess HyperSINDy's capacity for Bayesian inference/stochastic modeling for high dimensional stochastic systems, which are not amenable to existing analytical SDE discovery methods (e.g., (Boninsegna et al., 2018; Callaham et al., 2021)). Thus, we simulate a stochastic version of the Lorenz-96 system using:

$$\dot{x}_i = F_i + x_{i+1}x_{i-1} - x_{i-2}x_{i-1} - x_i \tag{11}$$

for $i = 1, ..., 10$ where $x_{-1} = x_9$, $x_0 = x_{10}$, and $x_{11} = x_1$. We iteratively sample each $F_i$ from a normal distribution: $F_i \sim \mathcal{N}(8, 10)$. As shown in Fig. 4, HyperSINDy correctly identifies all terms in the system, while also correctly learning a high variance coefficient exclusively for the forcing terms, $F_i$. In addition, HyperSINDy produces sample trajectories that match the stochastic dynamical behavior of ground truth sample trajectories. Refer to appendix Table S5 for quantitative comparisons between HyperSINDy and E-SINDy on the 10D Lorenz-96 system.

## 5 DISCUSSION

We have provided an overview of HyperSINDy, a neural network-based approach to sparse equation discovery for stochastic dynamics. HyperSINDy is unique in its ability to provide analytical representations and UQ in the setting of high-dimensional stochastic dynamics. The present work represents a proof of concept for this architecture. We envision numerous future directions for extending the algorithmic and theoretical aspects of HyperSINDy – e.g., evaluation in the context of other noise types and with respect to convergence in the continuous limit. Moreover, while we employ a fairly straightforward implementation of SINDy, numerous developments of the SINDy framework (Kaptanoglu et al., 2022) may be easily incorporated into the HyperSINDy architecture. Finally, the integration of SINDy into a neural network framework paves the way for future developments that incorporate advances in probabilistic machine learning with interpretable equation discovery.

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

## APPENDIX A    DERIVATION OF LOSS FUNCTION

We assume independence of $\mathbf{z}$ with respect to $\mathbf{x}$, such that $p_\theta(\mathbf{z}|\mathbf{x}) = p_\theta(\mathbf{z})$. Then, as described in the Methods section, our generative model factorizes as follows (Bayes' rule):

$$p_\theta(\dot{\mathbf{x}}, \mathbf{z}|\mathbf{x}) = p_\theta(\dot{\mathbf{x}}|\mathbf{z}, \mathbf{x})p_\theta(\mathbf{z}) \tag{12}$$

Given the chain rule for conditional probability, we also have:

$$
\begin{aligned}
p_\theta(\dot{\mathbf{x}}|\mathbf{z}, \mathbf{x}) &= \frac{p_\theta(\dot{\mathbf{x}}, \mathbf{z}, \mathbf{x})}{p_\theta(\mathbf{z}, \mathbf{x})} && \text{Conditional Probability} \\
&= \frac{p_\theta(\mathbf{z}|\dot{\mathbf{x}}, \mathbf{x})p_\theta(\dot{\mathbf{x}}|\mathbf{x})p_\theta(\mathbf{x})}{p_\theta(\mathbf{z}|\mathbf{x})p_\theta(\mathbf{x})} \\
&= \frac{p_\theta(\mathbf{z}|\dot{\mathbf{x}}, \mathbf{x})p_\theta(\dot{\mathbf{x}}|\mathbf{x})}{p_\theta(\mathbf{z})} && p_\theta(\mathbf{z}|\mathbf{x}) = p_\theta(\mathbf{z})
\end{aligned}
$$

By substituting into the first factorization, we obtain the second factorization:

$$p_\theta(\dot{\mathbf{x}}, \mathbf{z}|\mathbf{x}) = p_\theta(\mathbf{z}|\dot{\mathbf{x}}, \mathbf{x})p_\theta(\dot{\mathbf{x}}|\mathbf{x}) \tag{13}$$

We seek to learn a model that captures the dynamics $\dot{\mathbf{x}}$, given the state $\mathbf{x}$. Specifically, we seek to maximize the log-likelihood $\log p_\theta(\dot{\mathbf{x}}|\mathbf{x})$ by performing inference over $\mathbf{z}$. We follow a similar derivation as in (Kingma & Welling, 2019):

$$
\begin{aligned}
\log p_\theta(\dot{\mathbf{x}}|\mathbf{x}) &= \mathbb{E}_{q_\phi(\mathbf{z}|\dot{\mathbf{x}}, \mathbf{x})}\left[\log p_\theta(\dot{\mathbf{x}}|\mathbf{x})\right] \\
&= \mathbb{E}_{q_\phi(\mathbf{z}|\dot{\mathbf{x}}, \mathbf{x})}\left[\log \frac{p_\theta(\dot{\mathbf{x}}, \mathbf{z}|\mathbf{x})}{p_\theta(\mathbf{z}|\dot{\mathbf{x}}, \mathbf{x})}\right] && \text{see Eq. 13} \\
&= \mathbb{E}_{q_\phi(\mathbf{z}|\dot{\mathbf{x}}, \mathbf{x})}\left[\log \frac{p_\theta(\dot{\mathbf{x}}, \mathbf{z}|\mathbf{x})q_\phi(\mathbf{z}|\dot{\mathbf{x}}, \mathbf{x})}{p_\theta(\mathbf{z}|\dot{\mathbf{x}}, \mathbf{x})q_\phi(\mathbf{z}|\dot{\mathbf{x}}, \mathbf{x})}\right] \\
&= \mathbb{E}_{q_\phi(\mathbf{z}|\dot{\mathbf{x}}, \mathbf{x})}\left[\log \frac{p_\theta(\dot{\mathbf{x}}, \mathbf{z}|\mathbf{x})}{q_\phi(\mathbf{z}|\dot{\mathbf{x}}, \mathbf{x})}\right] + \mathbb{E}_{q_\phi(\mathbf{z}|\dot{\mathbf{x}}, \mathbf{x})}\left[\log \frac{q_\phi(\mathbf{z}|\dot{\mathbf{x}}, \mathbf{x})}{p_\theta(\mathbf{z}|\dot{\mathbf{x}}, \mathbf{x})}\right] \\
&= ELBO + D_{KL}(q_\phi(\mathbf{z}|\dot{\mathbf{x}}, \mathbf{x})||p_\theta(\mathbf{z}|\dot{\mathbf{x}}, \mathbf{x}))
\end{aligned}
$$

Since $D_{KL}(q_\phi(\mathbf{z}|\dot{\mathbf{x}}, \mathbf{x})||p_\theta(\mathbf{z}|\dot{\mathbf{x}}, \mathbf{x})) \geq 0$, we maximize the $ELBO$, which lower bounds $\log p_\theta(\dot{\mathbf{x}}|\mathbf{x})$.

$$
\begin{aligned}
ELBO &= \mathbb{E}_{q_\phi(\mathbf{z}|\dot{\mathbf{x}}, \mathbf{x})}\left[\log \frac{p_\theta(\dot{\mathbf{x}}, \mathbf{z}|\mathbf{x})}{q_\phi(\mathbf{z}|\dot{\mathbf{x}}, \mathbf{x})}\right] \\
&= \mathbb{E}_{q_\phi(\mathbf{z}|\dot{\mathbf{x}}, \mathbf{x})}\left[\log p_\theta(\dot{\mathbf{x}}, \mathbf{z}|\mathbf{x}) - \log q_\phi(\mathbf{z}|\dot{\mathbf{x}}, \mathbf{x})\right] \\
&= \mathbb{E}_{q_\phi(\mathbf{z}|\dot{\mathbf{x}}, \mathbf{x})}\left[\log p_\theta(\dot{\mathbf{x}}|\mathbf{z}, \mathbf{x}) + \log p_\theta(\mathbf{z}) - \log q_\phi(\mathbf{z}|\dot{\mathbf{x}}, \mathbf{x})\right] && \text{see Eq. 12} \\
&= \mathbb{E}_{q_\phi(\mathbf{z}|\dot{\mathbf{x}}, \mathbf{x})}\left[\log p_\theta(\dot{\mathbf{x}}|\mathbf{z}, \mathbf{x}) - D_{KL}(q_\phi(\mathbf{z}|\dot{\mathbf{x}}, \mathbf{x})||p_\theta(\mathbf{z}))\right]
\end{aligned}
$$

Equivalently, we can minimize the $-ELBO$, given by the following loss:

$$loss = (\dot{\mathbf{x}} - f_{\hat{\mathbf{z}}}(\mathbf{x}))^2 + D_{KL}(q_\phi(\mathbf{z}|\mathbf{x}, \dot{\mathbf{x}})||p_\theta(\mathbf{z}))$$

Given our goal to learn a sparse set of governing equations, we need to train $M$, which is multiplied elementwise with $\Xi_z$. To do so, we add $L_0(M)$ (main text Eq. 5) to the loss, yielding:

$$loss = (\dot{\mathbf{x}} - f_{\hat{\mathbf{z}}}(\mathbf{x}))^2 + \beta D_{KL}(q_\phi(\mathbf{z}|\mathbf{x}, \dot{\mathbf{x}})||p_\theta(\mathbf{z})) + \lambda L_0(M)$$

$$= (\dot{\mathbf{x}} - f_{\hat{\mathbf{z}}}(\mathbf{x}))^2 + \beta D_{KL}(q_\phi(\mathbf{z}|\mathbf{x}, \dot{\mathbf{x}})||p_\theta(\mathbf{z})) + \lambda \sum_{j=1}^{k} \text{Sigmoid}(\log \alpha_j - \beta_{L_0} \log \frac{\gamma}{\zeta})$$

where $\beta$, $\lambda$, $\beta_{L_0}$, $\gamma$, and $\zeta$ are hyperparameters, $k$ is the dimension of a vectorized $M$, and $\hat{\mathbf{z}} \sim q_\phi(\mathbf{z}|\mathbf{x}, \dot{\mathbf{x}})$ using the reparameterization trick. $log\alpha_j$ are location parameters for the distribution that M is transformed from, as described in the Background section of the main text.

Table S1: Matrix shapes for different experiments

| System | n | C | $\Theta(x)$ | $\Xi_z$ | $M$ | $\Theta(x)(\Xi_z \odot M)$ |
|---|---|---|---|---|---|---|
| Lorenz | 3 | F | $250 \times 19$ | $250 \times 19 \times 3$ | $250 \times 19 \times 3$ | $250 \times 3$ |
| Rössler | 3 | T | $250 \times 20$ | $250 \times 20 \times 3$ | $250 \times 20 \times 3$ | $250 \times 3$ |
| Lotka-Volterra | 2 | T | $250 \times 10$ | $250 \times 10 \times 2$ | $250 \times 10 \times 2$ | $250 \times 2$ |
| Lorenz-96 | 10 | T | $250 \times 286$ | $250 \times 286 \times 10$ | $250 \times 286 \times 10$ | $250 \times 10$ |

Table S2: Dataset initial conditions

| System | Train | Test |
|---|---|---|
| Lorenz | (0, 1, 1.05) | (-1, 2, 0.5) |
| Rössler | (0, 1, 1.05) | (-1, 2, 0.5) |
| Lotka-Volterra | (4, 2) | (2.1, 1.0) |
| Lorenz-96 | (8.01, 8, 8, 8, 8, 8, 8, 8, 8, 8) | (7.8, 8.7, 8.5, 6.0, 9.9, 9.5, 7.5, 6.9, 6.9, 8.7) |

## APPENDIX B  GENERATIVE AND INFERENCE MODELS

Matrix dimensions are variable. Consider $x \in \mathbb{R}^n$ and a library with $l$ terms. Then, we have:

$$\Theta(x) \in \mathbb{R}^l \quad \Xi_z \in \mathbb{R}^{l \times n} \quad M \in \mathbb{R}^{l \times n} \quad (\Xi_z \odot M) \in \mathbb{R}^{l \times n} \quad \Theta(x)(\Xi_z \odot M) \in \mathbb{R}^n$$

However, we use minibatches during training. Consider a batch of $x$ of size $b$, meaning $x \in \mathbb{R}^{b \times n}$. Then, we have:

$$\Theta(x) \in \mathbb{R}^{b \times l} \quad \Xi_z \in \mathbb{R}^{b \times l \times n} \quad M \in \mathbb{R}^{b \times l \times n} \quad (\Xi_z \odot M) \in \mathbb{R}^{b \times l \times n} \quad \Theta(x)(\Xi_z \odot M) \in \mathbb{R}^{b \times n}$$

After training, we do not sample $M$ using the reparameterization trick, since $\alpha$ has been learned. So, $M$ has shape $l \times n$ (note that $k = l \cdot n$). For all experiments, we included polynomials up to order 3 in the library and used a batch size of 250 during training. Refer to Table $S1$ for a breakdown of matrix shapes for each experiment during training (note that $C$ refers to whether a constant is included in the library).

## APPENDIX C  DATA

Each trajectory in the main manuscript was generated for 10000 timesteps with $\Delta t = 0.01$. Refer to Table S2 for data generation initial conditions. Note that the test initial condition for Lorenz-96 is rounded (the exact values can be found in the accompanying code, as we used Gaussian noise to choose the initial condition). Derivatives are estimated using finite differences without smoothing.

## APPENDIX D  TRAINING

### D.1  ALGORITHM

A "best practice" for the HyperSINDy training algorithm is choosing low initial $\beta$ and $\lambda$ values and evaluating the results before adjusting in future runs. Moreover, we also utilize beta warmup Castrejon et al. (2019), which is useful for avoiding posterior collapse in VAEs. Specifically, we increase the $\beta$ value from 0.01 to the chosen low initial $\beta$ value over 100 epochs. If the prior did not learn the function well enough, we increased ("spiked") the $\beta$ value at a later epoch in training to $\beta_{spike}$. Note that, although we knew the ground truth coefficients in our simulations, one can determine whether the prior learned "well enough" by comparing the similarity between the coefficients generated from the prior to the coefficients generated from the approximate posterior. See (Yacoby et al., 2022) for more information on this tradeoff between the posterior and prior. If the learned model was not sparse enough, we increased the $\lambda$ value at a later epoch in training to $\lambda_{spike}$.

Every 100 epochs, we permanently set values in $M$ to be zero if the corresponding coefficients (using the mean over a batch of coefficients) falls below the threshold value $T$. This is done using an auxiliary

matrix of shape $l \times n$, where the values are all initially set to 1 and then set to 0 throughout training if the corresponding value in $M$ should be 0 permanently. This mask is multiplied elementwise with $M$ to enforce this permanent sparsity. During training, we sample one M for each example in a minibatch of data using the reparameterization trick.

## D.2  HYPERPARAMETERS

Hyperparamater tuning mostly consists of adjusting the $\beta$ and $\lambda$ value for the Kl divergence and $L_0$ terms in the loss function, respectively. Hyperparameters that stay constant for all experiments are listed here: $learning\_rate = 0.005$, $num\_hidden = 5$, $stat\_size = 250$, $batch\_size = 250$. $stat\_size$ refers to the number of coefficient matrices that are sampled from the prior to calculate the coefficient means used for the permanent thresholding described in the Training section. We used a hidden dimension of 64 in all neural networks for all experiments except on Lorenz-96, for which we used a hidden dimension of 128. We warm up to a low initial $\beta$ value of 10 for every experiment. We use an initial $L_0$ regularization weight of $\lambda = 0.01$. For $M$, we use $\beta_{L_0} = 2/3$, $\gamma = -0.1$, and $\zeta = 1.1$. Note that an exhaustive list of hyperparameters and training settings can be found in the accompanying code. All experiments were run on an NVIDIA GeForce RTX 2080 Ti GPU. We ran all experiments in PyTorch (Paszke et al., 2019) using the AdamW optimizer (Loshchilov & Hutter, 2019) with a weight decay value of 0.01 and amsgrad (Reddi et al., 2019) enabled. Refer to Table S3 for a list of hyperparameters that we tuned to obtain the results in the main text. Refer to the attached code for more details on the RMSE experiments, as well as information on settings used to generate the supplemental figures.

## APPENDIX E   RMSE, PRECISION, AND RECALL METRICS

We use the following RMSE, precision, and recall metrics (as computed in Sun et al. (2022)):

$$rmse = \frac{||\mathbf{C}_{True} - \mathbf{C}_{Pred}||_2}{||\mathbf{C}_{True}||_2}$$

$$precision = \frac{||\mathbf{C}_{True} \odot \mathbf{C}_{Pred}||_0}{||\mathbf{C}_{Pred}||_0}$$

$$recall = \frac{||\mathbf{C}_{True} \odot \mathbf{C}_{Pred}||_0}{||\mathbf{C}_{True}||_0}$$

where $\mathbf{C}_{True}$ is the true mean or standard deviation of a given coefficient, and $\mathbf{C}_{Pred}$ is the mean or standard deviation of the predicted coefficients. For terms that are not included in the ground truth or predicted equations, we consider their mean or standard deviation to be zero.

## APPENDIX F   RDE-SDE TRANSFORMATION

Any finite dimensional SDE can be transformed into an RDE (Imkeller & Schmalfuss, 2001; Han & Kloeden, 2017). For an SDE with additive noise, the standard procedure involves replacement of the white noise with a stationary Ornstein Uhlenbeck process. Thus, following Han & Kloeden (2017) (section 3.5), the SDE

$$\mathrm{d}X_t = f(X_t)\mathrm{d}t + \mathrm{d}W_t \tag{14}$$

becomes

$$\dot{Z}_t = f(Z_t + O_t) + O_t, \tag{15}$$

where $Z_t \coloneqq X_t - O_t$ and $O_t$ is the stationary Ornstein-Uhlenbeck process, which is the solution to the SDE $\mathrm{d}O_t = -O_t\mathrm{d}t + \mathrm{d}W_t$ (with $W_t$ denoting the Wiener process).

In the multiplicative case, again following Han & Kloeden (2017), the SDE:

Table S3: Hyperparameters

| | Lorenz | | | | | | |
|---|---|---|---|---|---|---|---|
| $\sigma$ | $d$ | $\beta_{spike}$ | $epoch_{\beta_{spike}}$ | $\lambda_{spike}$ | $epoch_{\lambda_{spike}}$ | $epochs$ | $T$ |
| 1 | 6 | 100 | 400 | 10 | 400 | 999 | 0.05 |
| 5 | 6 | 400 | 400 | 10 | 400 | 999 | 0.05 |
| 10 | 6 | 400 | 400 | 10 | 400 | 999 | 0.05 |
| | Rössler | | | | | | |
| $\sigma$ | $d$ | $\beta_{spike}$ | $epoch_{\beta_{spike}}$ | $\lambda_{spike}$ | $epoch_{\lambda_{spike}}$ | $epochs$ | $T$ |
| 1 | 6 | 100 | 200 | 0.1 | 200 | 499 | 0.01 |
| 5 | 6 | 100 | 200 | 0.1 | 300 | 600 | 0.01 |
| 10 | 6 | 100 | 200 | 1 | 300 | 600 | 0.01 |
| Lorenz RMSE | | | | | | | |
| $\sigma$ | $d$ | $\beta_{spike}$ | $epoch_{\beta_{spike}}$ | $\lambda_{spike}$ | $epoch_{\lambda_{spike}}$ | $epochs$ | $T$ |
| 1 | 6 | 100 | 400 | 10 | 400 | 999 | 0.05 |
| 5 | 6 | 400 | 400 | 10 | 400 | 999 | 0.05 |
| 10 | 6 | 400 | 400 | 10 | 400 | 999 | 0.05 |
| Rössler RMSE | | | | | | | |
| $\sigma$ | $d$ | $\beta_{spike}$ | $epoch_{\beta_{spike}}$ | $\lambda_{spike}$ | $epoch_{\lambda_{spike}}$ | $epochs$ | $T$ |
| 1 | 6 | 100 | 200 | 0.1 | 200 | 499 | 0.01 |
| 5 | 6 | 200 | 200 | 0.1 | 300 | 600 | 0.01 |
| 10 | 6 | 300 | 200 | 1 | 300 | 600 | 0.01 |
| Lotka-Volterra | | | | | | | |
| $\sigma$ | $d$ | $\beta_{spike}$ | $epoch_{\beta_{spike}}$ | $\lambda_{spike}$ | $epoch_{\lambda_{spike}}$ | $epochs$ | $T$ |
| N/A | 4 | None | None | 0.1 | 100 | 250 | 0.1 |
| | Lorenz-96 | | | | | | |
| $\sigma$ | $d$ | $\beta_{spike}$ | $epoch_{\beta_{spike}}$ | $\lambda_{spike}$ | $epoch_{\lambda_{spike}}$ | $epochs$ | $T$ |
| 10 | 20 | None | None | 10 | 400 | 999 | 0.05 |

$$\mathrm{d}X_t = f(t, X_t)\mathrm{d}t + b(t)X_t \mathrm{d}Wt \tag{16}$$

may be combined with the random transformation

$$z(t) = T(t)X_t, T(t) := \exp\left(\frac{1}{2}\int_0^t b^2(s)\,\mathrm{d}s - \int_0^t b(s)\,\mathrm{d}W_s\right) \tag{17}$$

to obtain the RDE:

$$\frac{\mathrm{d}z}{\mathrm{d}t} = T(t)f\left(t, T^{-1}(t)z(t)\right). \tag{18}$$

In general, explicit RDE-SDE transformations may not always be straightforward to implement (see Imkeller & Schmalfuss (2001); Caraballo & Han (2016); Han & Kloeden (2017) for extended discussion of this topic). The present manuscript does not seek to establish equivalence or explicit mappings between particular RDE and SDE expressions, nor are claims contingent on the ability to carry out this transformation. Rather, for our purposes, we simply note that the conjugacy between

the two formulations implies that results obtained within one framework are directly pertinent to the other Han & Kloeden (2017).

More generally, the chief motivation for stochastic equations (whether SDEs or RDEs) lies simply in the mathematical modeling of dynamics under uncertainty; for many (if not most) modeling applications, the precise physical nature of this uncertainty -– e.g., a diffusion process or fluctuating system parameters — is unknown and/or of secondary importance (Friedrich et al., 2011; Duan, 2015; Särkkä & Solin, 2019). From this perspective, SDEs and RDEs simply emerge as alternative modeling frameworks, each with unique practical (dis)advantages to be considered in context (e.g., see Bauer et al. (2017)). Nonetheless future work may examine the possibility of equipping our approach with an explicit SDE prior (e.g., (Solin et al., 2021)), thus strengthening theoretical connections to a broader stochastic dynamics literature.

## APPENDIX G    ALGORITHMS

---

**Algorithm 1** Generation of Governing Equations Coefficients, $\Xi_z$, to predict $\dot{\mathbf{x}}$

---

1: **if** $\mathbf{z}$ not given **then**:
2:     $\mathbf{z} \sim \mathcal{N}(0, I)$                                               ▷ Generate batch of $\mathbf{z}$
3: $\Xi_{\mathbf{z}} = H(\mathbf{z})$                          ▷ Generate 1 coefficient matrix for each $\mathbf{z}$ in batch
4: $\dot{\mathbf{x}} = f_{\mathbf{z}}(\mathbf{x}) = \Theta(\mathbf{x})(\Xi_{\mathbf{z}} \odot M)$               ▷ Unless training, uses Eq 4 to get M

---

---

**Algorithm 2** Training Loop for Each Epoch

---

1: **for** each minibatch $\mathbf{x}, \dot{\mathbf{x}}$ **do**
2:     $\mu_q, \sigma_q = E(\mathbf{x}, \dot{\mathbf{x}})$                                  ▷ Encode each element of batch
3:     $\hat{\mathbf{z}} = \mu_q + \sigma_q \odot \epsilon$                                    ▷ Reparameterization Trick
4:     Obtain training $M$ through transformations         ▷ See Background, specifically Eq 3
5:     $\hat{\mathbf{x}} = f_{\hat{\mathbf{z}}}(\mathbf{x}) = $ Algorithm 1($\hat{\mathbf{z}}$)                        ▷ Give $\hat{\mathbf{z}}$ to H
6:     $loss = (\dot{\mathbf{x}} - \hat{\dot{\mathbf{x}}})^2 + \beta D_{KL}(q_\phi(\mathbf{z}|\mathbf{x}, \dot{\mathbf{x}})||p_\theta(\mathbf{z})) + \lambda L_0(M)$
7:     Backprop $loss$ and update $\theta, \phi$
8: Sample batch of $\mathbf{z} \sim \mathcal{N}(0, 1)$
9: $\Xi_{\mathbf{z}} = H(\mathbf{z})$
10: $\Xi_{\mathbf{z}_{mean}} = $ mean over batch of $\Xi_{\mathbf{z}}$
11: **if** (epoch % threshold_interval) == 0 **then**:                       ▷ If we must threshold this epoch
12:     $M = 0$ permanently where $abs(\Xi_{\mathbf{z}_{mean}}) < threshold$               ▷ Permanently threshold M

---

## APPENDIX H    FURTHER SIMULATION DETAILS

To generate results for Figure 1 and Table S4, we ran separate experiments generating 10 trajectories (each with a different random seed, and each generated from a different initial condition) for each noise level of both systems. In total, we trained one HyperSINDy model and one E-SINDy model on each trajectory, yielding 30 HyperSINDy models and 30 E-SINDy models.

## APPENDIX I    FIGURES AND TABLES

We include here additional sample trajectories (all from the test initial condition) to highlight HyperSINDy's generative capabilities. Refer to Figure S1 for HyperSINDy trajectories generated for various noise levels on the Lorenz and Rössler system, and refer to Figure S2 for sample test trajectories. HyperSINDy captures the same dynamical behavior as the original system. Refer to Figure S3 for HyperSINDy trajectories generated for the Lorenz-96 system. Refer to Figure S5 for results on a Lotka-Voltera system simulated as an RDE with half-normal noise on the coefficients. We compare the distribution of discovered HyperSINDy coefficients and E-SINDy coefficients with the ground truth coefficients.

Furthermore, we include numerous tables of quantitative comparisons between HyperSINDy, E-SINDy, and Bayesian Spline Learning (BSL) (Sun et al., 2022). Refer to Table S4 for precision and recall comparisons between HyperSINDy and E-SINDy on the Lorenz and Rössler systems (the experimental setup in this comparison is analogous to that of 1). Refer to Table S5 for comparisons between HyperSINDy and E-SINDy on Lorenz-96 systems. Refer to Table S6 for comparisons between HyperSINDy, E-SINDy, and BSL on a Lotka-Voltera system simulated as a RDE (i.e. with random gaussian noise on each coefficient, which differs from Figure 3, which used an SDE formulation).

Note that, in our experiments, we simulate a total of three types of Lotka-Voltera systems. In Figure 3, we simulate the Lotka-Voltera system as an SDE, using equation 10. As noted in the main text, we use $dt = 0.01$. In Figure S5, we simulate the Lotka-Voltera system as an RDE:

$$\dot{x} = \alpha_1 x - \alpha_2 xy \qquad\qquad \dot{y} = -\beta_1 y + \beta_2 xy \qquad (19)$$

At each timestep in the simulation, we draw samples of the coefficients from $HalfNormal$ (HN) distributions:

$$\{\alpha_1, \alpha_2, \beta_1, \beta_2\} \sim \{HN(1,5), HN(1,5), HN(1,5), HN(1,5)\}$$

Note that here, we use $dt = 0.005$. In Table S6, we again simulate the Lotka-Voltera system as an RDE using equation 19. However at each timestep in the simulation, we draw samples of the coefficients from Gaussian distributions:

$$\{\alpha_1, \alpha_2, \beta_1, \beta_2\} \sim \{\mathcal{N}(1,\sigma), \mathcal{N}(1,\sigma), \mathcal{N}(1,\sigma), \mathcal{N}(1,\sigma)\}$$

Here, we also use $dt = 0.005$.

Moreover, we attempted to fairly tune hyperparameters when training E-SINDy and BSL, using only the best model we could produce for comparisons. In the case of BSL, we tested out the following hyperparameters (refer to the publicly available BSL code): $lam = \{0.001, 0.0001, 0.000001\}, eta = \{0.05, 0.01, 0.025, 0.005, 0.0005\}, ADOlearningrate = \{0.05, 0.005\}$. Note that, for E-SINDy, HyperSINDy, and BSL, the strength of the sparsity parameter can significantly impact precision and recall, as it helps determines which terms get thresholded out.

Table S4: Total Term Precision and Recall relative to ground truth equations

| STD | | Lorenz | | Rössler | |
|---|---|---|---|---|---|
| | | HyperSINDy | E-SINDy | HyperSINDy | E-SINDy |
| 1 | Precision | $\mathbf{0.9857} \pm 0.0452$ | $0.6045 \pm 0.0274$ | $\mathbf{0.9375} \pm 0.0884$ | $0.7534 \pm 0.0850$ |
| | Recall | $0.8714 \pm 0.0452$ | $\mathbf{1.0000} \pm 0.0000$ | $0.9571 \pm 0.0690$ | $\mathbf{1.0000} \pm 0.0000$ |
| 5 | Precision | $\mathbf{0.9875} \pm 0.0395$ | $0.4588 \pm 0.0468$ | $\mathbf{0.9250} \pm 0.0645$ | $0.5502 \pm 0.0456$ |
| | Recall | $0.9429 \pm 0.0738$ | $\mathbf{1.0000} \pm 0.0000$ | $0.9857 \pm 0.0452$ | $\mathbf{1.0000} \pm 0.0000$ |
| 10 | Precision | $\mathbf{0.9875} \pm 0.0395$ | $0.6491 \pm 0.0268$ | $\mathbf{0.9260} \pm 0.0829$ | $0.4846 \pm 0.0699$ |
| | Recall | $0.9857 \pm 0.0452$ | $\mathbf{1.0000} \pm 0.0000$ | $0.9571 \pm 0.0690$ | $\mathbf{1.0000} \pm 0.0000$ |

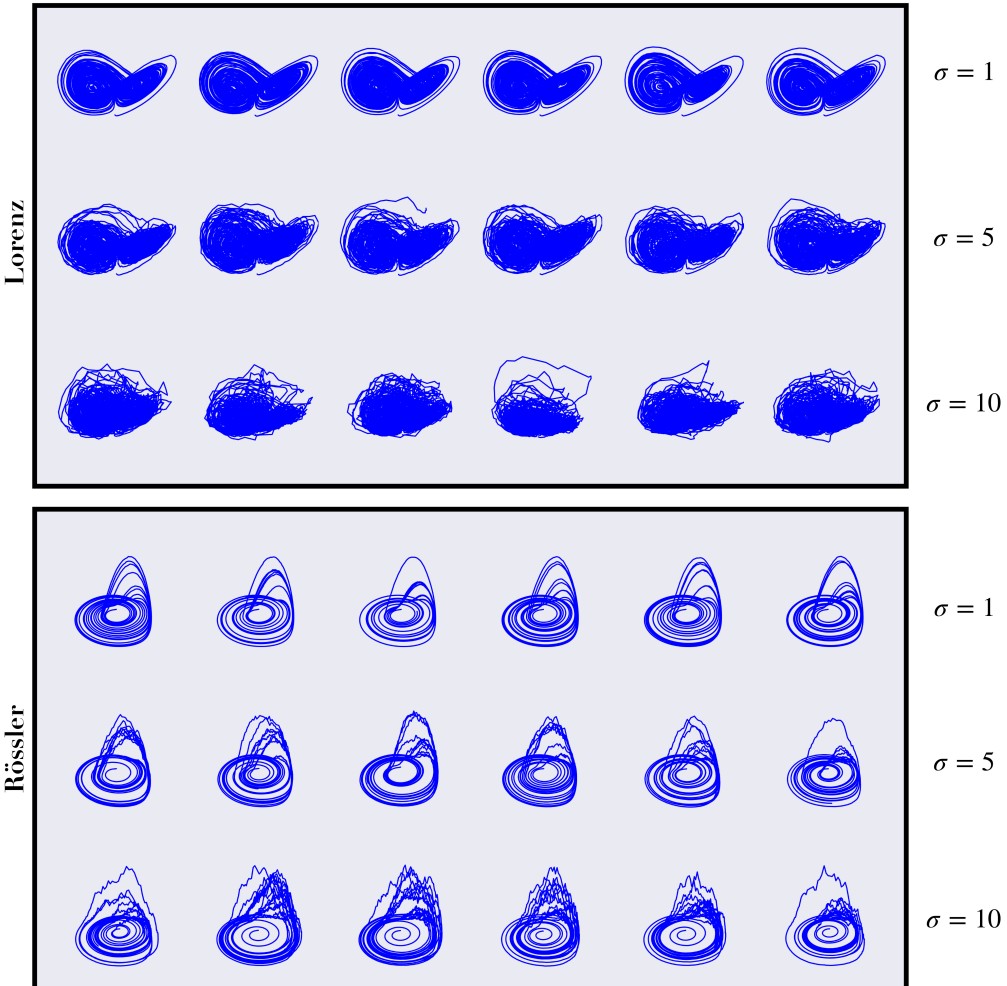

Figure S1: **Additional 3D Stochastic Lorenz and Rössler Samples**. HyperSINDy models trained on trajectories of varying noise ($\sigma$). Blue trajectories are generated by iteratively sampling from HyperSINDy's learned generative model.

Table S5: Lorenz-96

| Experiment | Method | RMSE Mean | RMSE STD | Precision | Recall |
|---|---|---|---|---|---|
| $\sigma = 0$ | HyperSINDy | 0.05227 | N/A | 1.0 | 1.0 |
| | E-SINDy | **0.006756** | N/A | 1.0 | 1.0 |
| $\sigma = 5$ | HyperSINDy | **0.05370** | **0.7375** | **1.0** | 1.0 |
| | E-SINDy | 0.1591 | 0.8240 | 0.4348 | 1.0 |
| $\sigma = 10$ | HyperSINDy | **0.1106** | **0.2117** | **1.0** | 1.0 |
| | E-SINDy | 0.1729 | 0.8544 | 0.3077 | 1.0 |

We simulate the Lorenz-96 with varying levels of Gaussian noise ($\sigma$) on the forcing term. We report the RMSE between the mean and standard deviation of discovered coefficients, as compared to ground truth; we also report precision and recall of the terms. Note that for $\sigma = 0$, we cannot report the RMSE of the learned STD, as the ground truth standard deviation is 0.

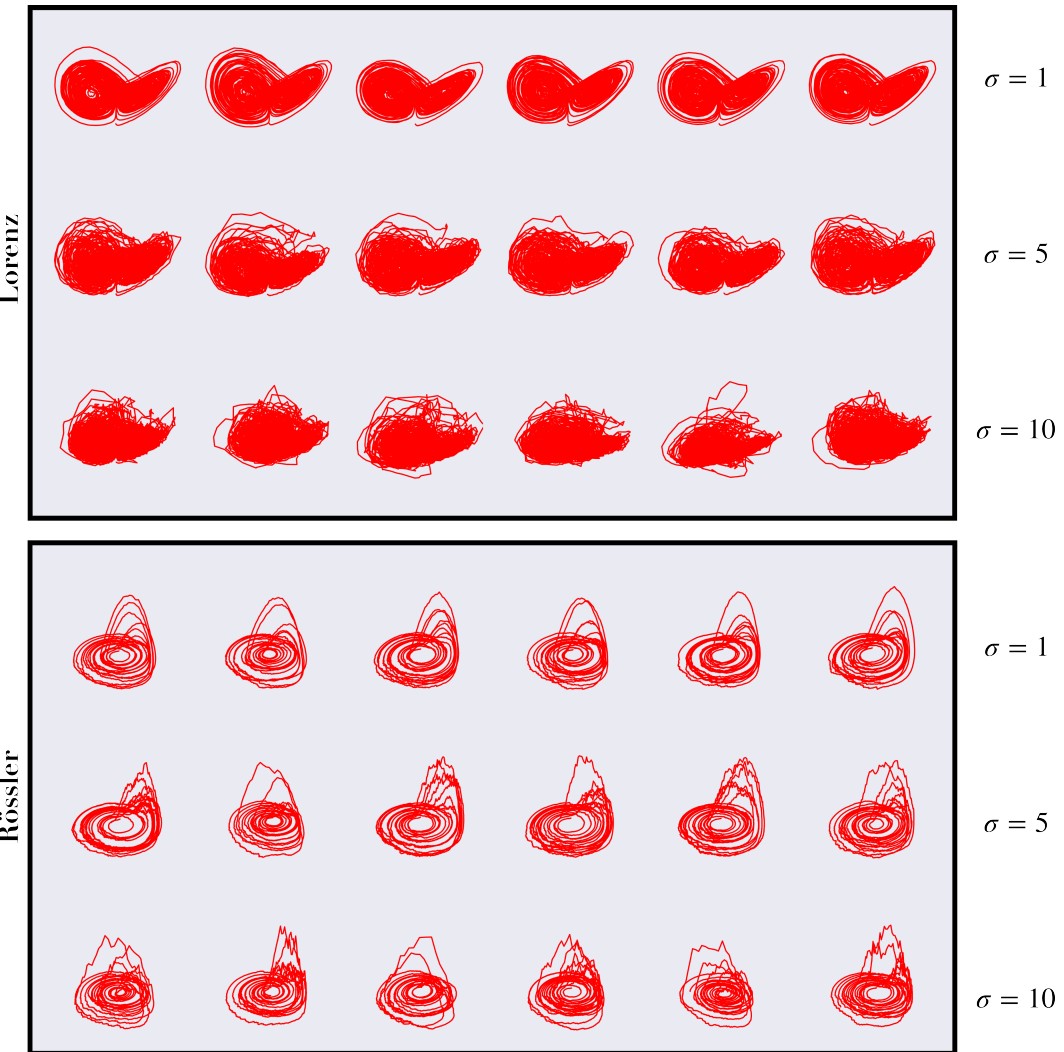

Figure S2: **3D Stochastic Lorenz and Rössler Ground Truth Samples**. Samples generated using the ground truth equations for varying noise levels ($\sigma$).

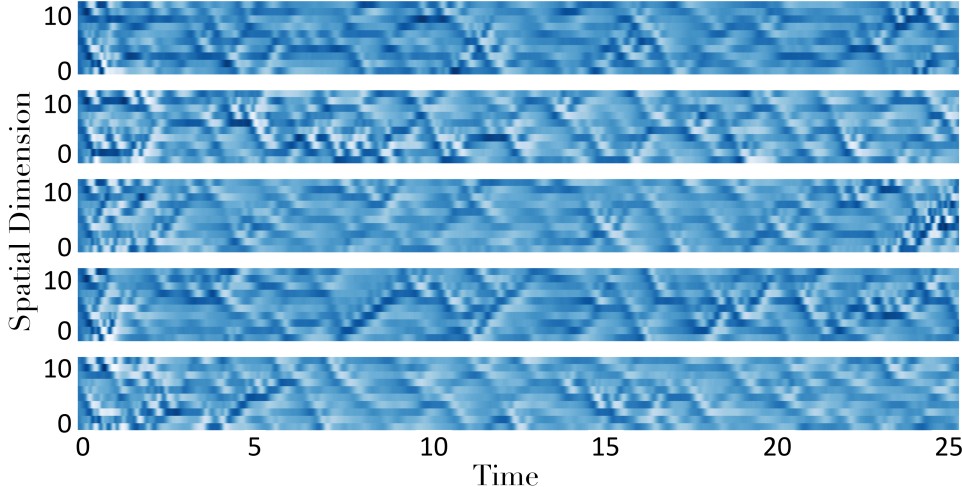

Figure S3: **Lorenz-96 Samples** ($\sigma = 10$). Each row contains a different sample trajectory. Trajectories are generated by iteratively sampling from HyperSINDy's learned generative model.

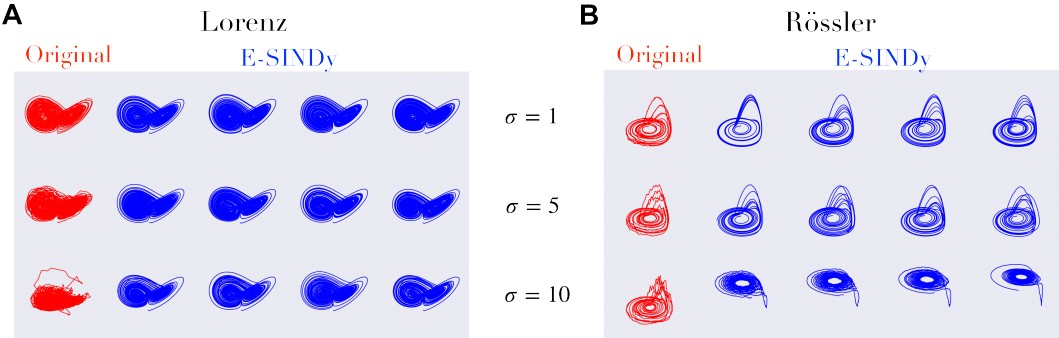

Figure S4: **E-SINDY 3D Stochastic Lorenz and Rössler Samples**. Samples generated using discovered E-SINDy equations for varying noise levels ($\sigma$).

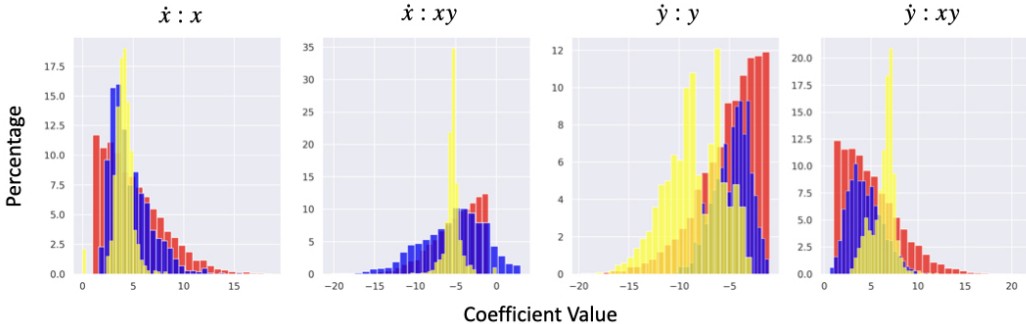

Figure S5: **Lotka-Volterra System with Half-Normal Noise** ($\sigma = 5$). Comparison of ground truth (red), HyperSINDy (blue) and E-SINDy (yellow) coefficient distributions for each of the dynamical terms. We simulate this system using an RDE formulation with half-normal noise (with the given $\sigma$) on each coefficient.

Table S6: Lotka-Volterra system with Gaussian noise on every coefficient.

| Experiment | Method | RMSE Mean | Precision | Recall |
|---|---|---|---|---|
| $\sigma = 0$ | HyperSINDy | 0.0028 | 1.0 | 1.0 |
| | E-SINDy | **0.0025** | 1.0 | 1.0 |
| | Bayesian Spline | 0.1870 | 1.0 | 1.0 |
| $\sigma = 1$ | HyperSINDy | **0.0415** | **1.0** | 1.0 |
| | E-SINDy | 0.1763 | 0.5714 | 1.0 |
| | Bayesian Spline | 0.2880 | 0.6667 | 1.0 |
| $\sigma = 2$ | HyperSINDy | **0.0902** | **1.0** | 1.0 |
| | E-SINDy | 0.9480 | 0.2857 | 1.0 |
| | Bayesian Spline | 0.4665 | 0.4444 | 1.0 |
| $\sigma = 3$ | HyperSINDy | **0.1694** | **0.8000** | 1.0 |
| | E-SINDy | 1.4953 | 0.3077 | 1.0 |
| | Bayesian Spline | 0.7577 | 0.4 | 1.0 |

We simulate the Lotka-Volterra system as an RDE, i.e. with Gaussian noise (with the given $\sigma$) on each coefficient. For each $\sigma$, we train a *HyperSINDy* model with the given $z$ dimension, then evaluate the RMSE of the mean and standard deviation of the discovered coefficients, as compared to the ground truth mean and standard deviation.

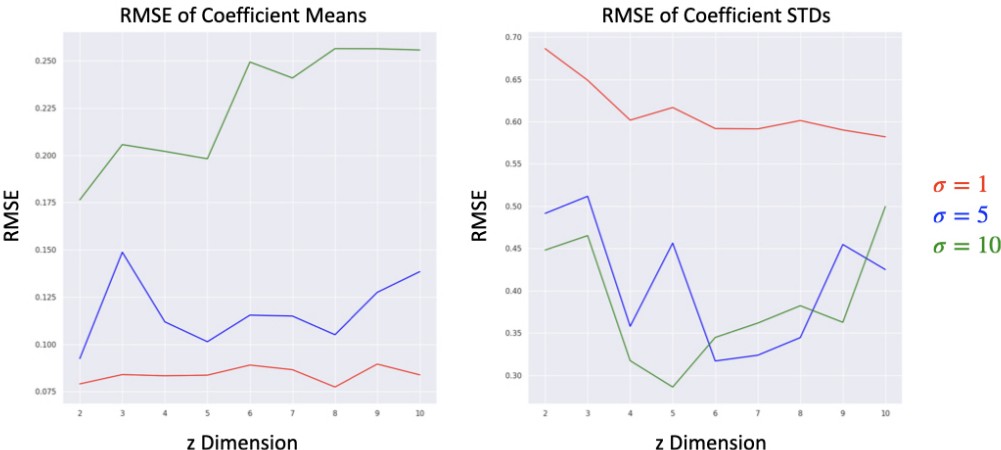

Figure S6: **3D Stochastic Lorenz with varying z dimension**. We simulate the 3D Lorenz system, as in 2, then train HyperSINDy models with different $z$ dimension on each trajectory. Here, we plot the RMSE of the discovered mean and standard deviation of coefficients, as compared to the ground truth. Note that, for each $\sigma$, even though the RMSE can vary significantly for different $z$ dimensions, it is still always lower than E-SINDy (see Table 1).

