# OpenReview forum: "HyperSINDy: Deep Generative Modeling of Nonlinear Stochastic Governing Equations"
_ICLR.cc/2024/Conference — Submitted to ICLR 2024_

### Official Review · Reviewer_Bwz8 · 2023-11-01

**Soundness:** 3 good
**Presentation:** 3 good
**Contribution:** 3 good
**Rating:** 6
**Confidence:** 5

**Summary:**

The author proposed HyperSINDy, a variant of SINDy for discovering stochastic dynamical systems. It employs a variational encoder and a sparsity promoting loss function to recover the underlying equation forms.

**Strengths:**

Originality: The author proposed a new learning framework by combining the auto-encoder and sparse-promoting loss functions for equation discovery tasks.

Quality: The result for the proposed method is comprehensive and includes various ODE cases.

Clarity: The figure and equation are well formulated in general. The writing is self-contained to understand

Significance:  The proposed methods integrates sparse equation discovery methods deep generative modeling.

**Weaknesses:**

The problem setting is confusing and the baseline models are not adequate. Details see the questions part.

**Questions:**

1. Problem setting about the stochastic system. As the author mentioned, they used an alternative definition of the stochastic dynamics, e.g random ODE. However, the reviewer is confused by the benefit of such definition compared to the deterministic setting. In result part 4.1, both the mean and the STD of the underlying system are shown. However, the STD form doesn't corresponds to any dynamics and the mean is close the the true mean of the system. In such circumstances, it seems only the mean estimation is important and the std cannot be leveraged to judge the performance of the proposed model. For part 4.2, the STD result is also confusing. It is compared with the diffusion terms but it is totally different from the true diffusion term. Therefore, the reviewer wants to ask why we need to include the STD and how it can help the proposed model.

2. The result needs to compare with more SOTA models and include more comprehensive metrics. There are several model combining learning based method and SINDy-like algorithm for the equation discovery tasks [1][2]. Also, by checking the Figure 2, we could find that the discovery form is different under different $\sigma$. For a equation discovery problem, it is important to get a consistent and correct form. Therefore the reviewer suggests adding more metrics on evaluating if the proposed model can get the correct form, e.g, precision and recall metrics.

[1] Bayesian Spline Learning for Equation Discovery of Nonlinear Dynamics with Quantified Uncertainty.

[2] Physics-informed learning of governing equations from scarce data

3. The high dimensional 10d lorenz 96 is not compared with any baselines. Moreover, the analytical form is not listed in the main manuscript. Figure 4's caption says check equation 9 but that's for lorenz 63. Lorenz 96 should like equation 12 but with the concrete forcing terms. Equation 12 indicates that all the coefficients except the forcing terms should be close to 1 or -1. However, the discovered coefficients for $x_i$ is not close to these values. Again, the reviewer doesn't understand what is the gain of reporting the STD form of the equation here.

4. The methodology part is confusing. Figures 2 says $\theta$ has 3 terms but page 8 says $\theta$ has 2 terms. Moreover, in the lower part of Figure 2, $z$ is firstly sampled from $p_{\theta}(z)$ then was fed into decoder $H$. However, the definition of $\theta$ has already included $H$, making the $H$ applied to $z$ 2 times. The term "inference model" is commonly used for test time, but the author use it to indicate training procedure. There are all the confusing parts need to be clarified.

---

> ### Author Response · Authors · 2023-11-19
> **Rebuttal (Part 1/2)**
>
> ### Weaknesses
>
> > 1. The problem setting is confusing and the baseline models are not adequate. Details see the questions part.
>
> **Response:** Please see responses below.
>
> ### Questions
> > 1. Problem setting about the stochastic system. As the author mentioned, they used an alternative definition of the stochastic dynamics, e.g random ODE. However, the reviewer is confused by the benefit of such definition compared to the deterministic setting. In result part 4.1, both the mean and the STD of the underlying system are shown. However, the STD form doesn't corresponds to any dynamics and the mean is close the the true mean of the system. In such circumstances, it seems only the mean estimation is important and the std cannot be leveraged to judge the performance of the proposed model. For part 4.2, the STD result is also confusing. It is compared with the diffusion terms but it is totally different from the true diffusion term. Therefore, the reviewer wants to ask why we need to include the STD and how it can help the proposed model.
>
> **Response:** We apologize for this confusion, which we believe stems from the interpretation of the RDE used to simulate the dynamics. In each case, mean and STD refer to the parameters of the (Gaussian) distribution associated with each coefficient; trajectories are simulated by constructing sample paths from these distributions. In other words, each coefficient is effectively coupled to an independent Wiener process; the system is parameterized by an n-dimensional Wiener  process. Accordingly, each ground truth trajectory is associated with only one particular sample path for the n-dimensional Wiener process.
>
> The STDs factor into the dynamics by specifying the level of dynamical noise that drives each coefficient. The relevance of the STD to the dynamics is most clear in the rows of Fig. 2 corresponding to $\sigma=10$. In these cases, the ground truth trajectory (“Original”) is highly stochastic. HyperSINDy succeeds in recovering the ground truth equation along with the correct coefficient means. However, the mean (middle column) is a poor representation of the original system, which behaves much more randomly. Thus, by discovering appropriately large STD values, HyperSINDy is able to accurately represent the level of uncertainty in the system’s evolution; this also enables HyperSINDy to faithfully recapitulate the dynamical behavior of the original system. E-SINDy fails to capture the appropriate level of uncertainty (as it assumes no dynamical noise); as such, it cannot faithfully recapitulate the dynamical behavior of the original system (Fig. S4).
>
> > 2. The result needs to compare with more SOTA models and include more comprehensive metrics. There are several model combining learning based method and SINDy-like algorithm for the equation discovery tasks [1][2]. Also, by checking the Figure 2, we could find that the discovery form is different under different sigma . For a equation discovery problem, it is important to get a consistent and correct form. Therefore the reviewer suggests adding more metrics on evaluating if the proposed model can get the correct form, e.g, precision and recall metrics.
>
> **Response:** We believe that this point is partially addressed by our clarification of the current problem setting as stochastic rather than merely noisy, deterministic dynamics. Nonetheless, we completely agree that comparison against another state-of-the-art method for probabilistic equation discovery, such as Bayesian Spline Learning, would improve the manuscript. We have updated the manuscript to include comparisons with this method along with additional quantitative metrics – thank you for these suggestions.
> Refer to Table S4 in the appendix for precision and recall results that complement the experiments ran for Table 1 in the main text (on Lorenz and Rossler systems generated with 10 seeds). Refer to Table S5 in the appendix for RMSE, precision, and recall comparisons between HyperSINDy and E-SINDy on multiple 10D Lorenz-96 trajectories, each generated with a different noise level. Refer to Table S6 in the appendix for RMSE, precision, and recall comparisons between HyperSINDy, E-SINDy, and Bayesian Spline Learning on multiple Lotka-Voltera trajectories simulated as RDEs with Gaussian noise on every coefficient. (response continued below)

---

> > ### Author Response · Authors · 2023-11-19
> > **Rebuttal (Part 2/2)**
> >
> > We agree that the manuscript can be further improved by including metrics to evaluate recovery of the proper equation form. We have added these metrics as suggested. We find that recall is generally high for all methods, while HyperSINDy generally yields significant improvement on the precision metrics. However, we note that both precision and recall may be more sensitive to specific parameter choices (especially concerning sparsity thresholds, as it helps determine how many terms get thresholded out).
> > Note that we were unsure how to adapt the publicly available code to cases other than 2-dimensional systems, so we did not evaluate Bayesian Spline Learning on our 10D Lorenz96 datasets. Instead, we report results for Bayesian Spline Learning on a 2D Lotka-Volterra system, simulated with the RDE formulation with random gaussian noise on every coefficient. Please refer to appendix Table S6 for these results.
> >
> > > 3. The high dimensional 10d lorenz 96 is not compared with any baselines. Moreover, the analytical form is not listed in the main manuscript. Figure 4's caption says check equation 9 but that's for lorenz 63. Lorenz 96 should like equation 12 but with the concrete forcing terms. Equation 12 indicates that all the coefficients except the forcing terms should be close to 1 or -1. However, the discovered coefficients for x_i is not close to these values. Again, the reviewer doesn't understand what is the gain of reporting the STD form of the equation here.
> >
> > **Response:** To the first point, we have run simulations comparing to E-SINDy; please refer to appendix Table S5 for these results. To the second point, we thank you for pointing out the error. Indeed, figure 4 should point to equation 12, not equation 9. We have updated the manuscript to reflect this change. To the third point, we agree that the coefficients should be close to 1 or -1. Most of the coefficients are close, except for the strictly x_i coefficients, as you point out. The main point of note is that the learned STD is appropriately scaled. The discovered STD reflects that of the underlying distribution, and notably only the learned forcing term has a high variance.
> >
> > > 4. The methodology part is confusing. Figures 2 says theta  has 3 terms but page 8 says  theta has 2 terms. Moreover, in the lower part of Figure 2, z is firstly sampled from p_theta(z),  then was fed into decoder . However, the definition of theta has already included H, making the H applied to z 2 times. The term "inference model" is commonly used for test time, but the author use it to indicate training procedure. There are all the confusing parts need to be clarified.
> >
> > **Response:**
> > We thank you for pointing out this confusion. In the case of the Lorenz and Rossler experiments, theta has 19 and 20 terms, respectively. What we display in figure 2 are the coefficients of the discovered equations, many of which only have 3 terms left due to sparsification. As for Page 8, we show RMSE over the mean and STD of all discovered coefficients, as compared to the ground truth (Table 1). Thus, for both mean and STD, we have one single mean and STD value (so 4 values in total). We have updated the manuscript to reflect changes to increase clarity – please let us know if anything remains unclear.
> > For our model, $\Theta$ (i.e., the function library) does not include H (the hypernetwork). $\Theta$ is a library constructed entirely using the state variables (“x” in our notation). Specifically, we construct $\Theta(x)$. Then, H takes in a latent vector (z) and outputs the coefficients, $\Xi_{z}$. We multiply $\Xi_{z}$ elementwise with the sparse mask M, then perform a matrix multiplication of the result with $\Theta(x)$ to predict the derivative.
> >
> > With regards to the term “inference model”, we use this term in accordance with standard VAE terminology to refer to a model that infers the latent vector from data observations (in our case, ($x,\dot{x}$). For HyperSINDy’s use case, this is exclusive to training, when (state, derivative) pairs are fed into the encoder to infer the latent vector, z. We do not use the term “inference model” to refer to the training procedure per se, but rather the model that is used to perform inference, which is utilized only during the training phase.
> >
> > Thank you for giving us the opportunity to clarify these points; please let us know if anything remains unclear.

---

> ### Comment · Reviewer_Bwz8 · 2023-11-23
> **Thanks for the detailed rebuttal**
>
> I have changed my score to reflect these efforts.

---

### Official Review · Reviewer_UKtm · 2023-11-01

**Soundness:** 3 good
**Presentation:** 3 good
**Contribution:** 3 good
**Rating:** 6
**Confidence:** 3

**Summary:**

This paper proposes a new framework HyperSINDy (Hyper sparse identification of nonlinear dynamics) to address the symbolic regression problems in high-dimensional, stochastic setting. Within a variational autoencoder, they use an encoder to learn the parameters $\mu, \sigma$ of the latent states $\mathbf{z}$, and a generative model to learn $p(\dot{\mathbf{x}}|\mathbf{x}, \mathbf{z})$ where $\mathbf{\dot{x}}$ is parameterized by $f_\mathbf{z}(\mathbf{x})$. With proper choice of $f_\mathbf{z}(\mathbf{x})$, they build the relationship between derivatives and coefficients for addressing the task of SINDy.

**Strengths:**

This paper is well written.

The idea of mapping a high-dimensional, stochastic data to a low dimensional latent space and learning the coefficients through a hyper network which takes low-dimensional latent variables are novel.

**Weaknesses:**

The capacity of $\Theta(\mathbf{x})$ still holds as a constraint for the performance, especially in the high-dimensional setup. It would be great if the authors could discuss the impact of the $\Theta(\mathbf{x})$. For example, what would the performance be if certain symbolic terms (shown up in the true equations) are missing in the dictionary in $\Theta(\mathbf{x})$.

**Questions:**

Q1. What is the column of ''STD'' in Figure 2 showing? Are they showing the standard deviation of the estimates? If that is the case, plugging in the standard deviation as the coefficients in the equations are confusing.

Q2. It would be great if the authors could provide more evaluation metrics for generated trajectories. Metrics like Lyapunov exponents would be helpful to see how good the performance is.

Q3. How robust the performance would be across different choice of the dimension of $\mathbf{z}$?

---

> ### Author Response · Authors · 2023-11-19
>
> ### Weakness
>
> > 1. The capacity of (theta(x)  still holds as a constraint for the performance, especially in the high-dimensional setup. It would be great if the authors could discuss the impact of theta(x) . For example, what would the performance be if certain symbolic terms (shown up in the true equations) are missing in the dictionary theta(x).
>
> **Response:** We agree that the model is constrained by the capacity of the function library. This is true of most equation discovery methods. The polynomial basis has broad applicability; moreover, there is often a priori knowledge motivating choice of specific classes of basis functions. The SINDy framework has had widespread success with alternative libraries (e.g., Kaptanoglu et al., 2022). Our method is amenable to such alternative libraries, and thus retains the flexibility of the SINDy framework.
>
> ### Questions
>
> > 1. What is the column of ''STD'' in Figure 2 showing? Are they showing the standard deviation of the estimates? If that is the case, plugging in the standard deviation as the coefficients in the equations are confusing.
>
> **Response:** Thank you for pointing out this confusion. Correct, this figure is showing the standard deviation of the estimated coefficients, although these standard deviations correspond to the noise processes that contribute to the dynamics. We have updated the figure to clarify the interpretation of the standard deviations.
>
> > 2.  It would be great if the authors could provide more evaluation metrics for generated trajectories. Metrics like Lyapunov exponents would be helpful to see how good the performance is.
>
> **Response:** Thank you for this excellent suggestion. Although we were unable to carry out a systematic analysis of the Lyupanov exponents for generated trajectories, we have incorporated two additional metrics, precision and recall (Tables S4-S6). These additional metrics enable assessment of the ability to recover the correct terms of the original equation.
>
> > 3. How robust the performance would be across different choice of the dimension of z?
>
> **Response:** Thank you for raising this important question. We have carried out additional experiments to assess performance as a function of latent dimension. Specifically, we train HyperSINDy models with varying dimensions of z on each of the 3D Stochastic Lorenz trajectories ($\sigma = 1$, $\sigma = 5$, $\sigma = 10$).
>
> Results from this procedure are shown in Fig. S6. We plot the RMSE of the mean and standard deviation of discovered coefficients, as compared to ground truth. It is true RMSE can vary as a function of z dimension. It is noteworthy, however, that for each trajectory, RMSE is always lower than E-SINDy, regardless of z dimension (refer to Table 1 for E-SINDy RMSE). These results help validate our choice of z in the paper: we use 2x the state dimension for z, and this seems to have good performance at balancing RMSE of both the mean and standard deviation of discovered coefficients.
>
> References
>
> Kaptanoglu, A.A. et al. (2022) ‘PySINDy: A comprehensive Python package for robust sparse system identification’, Journal of Open Source Software, 7(69), p. 3994. Available at: https://doi.org/10.21105/joss.03994.

---

> > ### Comment · Reviewer_UKtm · 2023-11-23
> >
> > I want to thank the authors for the detailed response. I appreciate the additional experiments, understand the constraints of the function library (and also the possibility of alleviation through prior knowledge), and am also aware of the ongoing discussions.

---

### Official Review · Reviewer_cj4i · 2023-11-01

**Soundness:** 2 fair
**Presentation:** 3 good
**Contribution:** 2 fair
**Rating:** 6
**Confidence:** 3

**Summary:**

The paper proposes HyperSindy, which is a framework for modeling stochastic nonlinear dynamics. First a variational autoencoder is used to model the distribution of observed states and derivatives. Samples from the VAE are used with a hypernetwork to obtain the coefficients of the differential equations. These coefficients are combined with a function library to obtain the derivatives, allowing for the functional form of the equations to be learned. Experiments are conducted using simulated data, which show promising results.

**Strengths:**

The paper aims to learn both the parameters and the functional form of stochastic differential equations from data, which is a significant problem for scientific applications. The use of VAEs and hypernetworks for this problem is quite novel to my knowledge. The paper is well written and organized.

**Weaknesses:**

Experiments are conducted in simulated environments where the simulation parameters match to the modeling assumptions (mainly around Gaussianity). I would love to see more experiments confirming the applicability of the approach to broader problems, especially with real data.

**Questions:**

- As mentioned above, all the experiments are conducted using Gaussian distributions which match to the posterior distribution assumed for variational inference. Can authors comment on the limitations of these experiments?
- The approach aims to learn both the functional form and parameters of the differential equations. Even though I agree that this might help with interpretability, I worry that the identifiability issues might be prominent. Do the authors expect any identifiability problems?
- The promise of learning functional form is achieved through the function library. Are there any limitations of using such an approach?
- What are the limitations of using a Gaussian prior with diagonal covariance for the generative model?

---

> ### Author Response · Authors · 2023-11-19
>
> ### Weaknesses
>
> >1. Experiments are conducted in simulated environments where the simulation parameters match to the modeling assumptions (mainly around Gaussianity). I would love to see more experiments confirming the applicability of the approach to broader problems, especially with real data.
>
> **Response:** The insight exploited in this paper is that the standard VAE gaussian prior is well-matched to the well-studied case of stochastic dynamics involving white noise. We completely agree that extension to other kinds of noise will be an important direction for future extensions. As an initial investigation into the feasibility for such extensions, we have included a new experiment (Fig. S5) that supports the potential for HyperSINDy to model more complicated noise types (here, noise drawn from a half-Normal distribution).
>
> ### Questions
>
> > 1. As mentioned above, all the experiments are conducted using Gaussian distributions which match to the posterior distribution assumed for variational inference. Can authors comment on the limitations of these experiments?
>
> **Response:** (see above response)
>
> > 2. The approach aims to learn both the functional form and parameters of the differential equations. Even though I agree that this might help with interpretability, I worry that the identifiability issues might be prominent. Do the authors expect any identifiability problems?
>
> > 3. The promise of learning functional form is achieved through the function library. Are there any limitations of using such an approach?
>
> **Response:** We agree that there can be identifiability issues in equation learning; addressing these issues generally comes at the cost of reduced flexibility. This is precisely the tradeoff reflected in these questions. Thus, specifying a function library and imposing sparsity are significant aids toward identifying a parsimonious model. Moreover, by assuming the structure of the noise, we avoid the major identifiability issues that arise when attempting to learn the form of both the deterministic and stochastic terms. These constraints each necessitate a loss of generality, though they are appealing as they retain very broad applicability.
> The primary limitation of a function library is that it requires a priori knowledge about the dynamics, i.e. some inductive bias is necessary. However, this inductive bias is also the primary motivation behind equation discovery methods: we want to discover a sparse equation from a finite set of possible terms that could describe the system’s dynamics. One alternative might be to enable a more fully data-driven search for the basis functions (e.g., use via a genetic algorithm (Schmidt and Lipson, 2009)), although such approaches are generally incur much greater computational cost.
>
> > 4. What are the limitations of using a Gaussian prior with diagonal covariance for the generative model?
>
> **Response:** One limitation is that such a prior assumes the true latent vector is built of multiple  i.i.d. random variables (diagonal covariance). However, one could certainly imagine noise processes for which this assumption does not hold. Even though the hypernetwork contains nonlinearities (and thus can model dependencies between coefficients), some systems may still be challenging to learn. Future work could explore using alternate priors, such as Gaussian Process Variational Autoencoder (Casale et al., 2018), which may better account for such dependencies.
>
> Despite this limitation, we note that the network is able to learn more complex distributions that include dependencies between coefficients. In particular, refer to Figure S5 in the appendix for results on a Lotka-Volterra system simulated with coefficients drawn from a Half-Normal distribution. Even though HyperSINDy uses a standard gaussian prior with diagonal covariance, it is able to approximate the shape of the distribution of the true non-Gaussian coefficients much better than E-SINDy. We view this experiment as a proof of principle for the flexibility provided by hypernetwork’s nonlinearities: it is able to learn a mapping from latent space (a simple standard gaussian) to a much more complex coefficient space.
>
> References
>
> Schmidt, M. and Lipson, H. (2009) ‘Distilling Free-Form Natural Laws from Experimental Data’, Science, 324(5923), pp. 81–85. Available at: https://doi.org/10.1126/science.1165893.
>
> Casale, F.P. et al. (2018) ‘Gaussian Process Prior Variational Autoencoders’, in Advances in Neural Information Processing Systems. Curran Associates, Inc. Available at: https://proceedings.neurips.cc/paper_files/paper/2018/hash/1c336b8080f82bcc2cd2499b4c57261d-Abstract.html (Accessed: 19 November 2023).

---

> > ### Comment · Reviewer_WJp7 · 2023-11-20
> >
> > I want to first thank the response and revision from the authors. However, I feel that the current manuscript still lacks clarification of some critical points that affect the quality of the proposed scheme.
> >
> > (1) As mentioned by the authors, it is in general not straightforward to transform a SDE to a RODE. In fact, the transformation could be highly non-trivial and limits the application of the proposed scheme. See for example "The shifted ODE method for underdamped Langevin MCMC. by James Foster, Terry Lyons and Harald Oberhauser".
> > (2) It is still not clear to me how to transform multiplicative noises.

---

> ### Author Response · Authors · 2023-11-20
>
> Thank you for your prompt follow-up. As elaborated in our latest response to Reviewer 7UM6, we stress that our manuscript does not claim the ability to explicitly map between particular RDE and SDE expressions, and our results are not contingent on the ability to carry out this transformation. Rather, we simply note that the conjugacy between the two formulations means that the results obtained in one framework are recognized to be directly pertinent to those obtained in the other (Han and Kloeden, 2017). Thus, our work retains relevance to a very broad class of problems that have been modeled as SDEs – ultimately, SDEs and RDEs are simply two alternative modeling choices for studying dynamics whose evolution involves some uncertainty.
>
> Note that the Lotka-Volterra SDE problem (Fig. 3) demonstrates the utility of our approach for a problem explicitly formulated as an SDE -- HyperSINDy captures the appropriate dynamics (as confirmed by evaluating the Kramers-Moyal coefficients) and, importantly, identifies the physical terms that are present in the drift component of the original SDE, thus supporting insights into the physical interactions in the system.
>
> We will update the Appendix to include these clarifications, as well as an example transformation for multiplicative noises (see latest response to Reviewer 7UM6.

---

### Official Review · Reviewer_kqcu · 2023-11-03

**Soundness:** 4 excellent
**Presentation:** 4 excellent
**Contribution:** 4 excellent
**Rating:** 8
**Confidence:** 3

**Summary:**

This work proposes HyperSINDy, a method for unsupervised discovery of governing differential equations from data in the stochastic setting. HyperSINDy combines variational Bayesian inference and hypernetworks (Ha et al., 2016) to build a generative model of the data. An L0 regularization scheme based on concrete random variables is used to ensure that the final differential equation learned is sparse and interpretable. HyperSINDy outperforms the previous state of the art in both random differential equation and stochastic differential equation settings.

**Strengths:**

- **The paper is very well written.** The presentation on the backgrounds (the SINDy framework, variational inference, L0 regularization) is very clear and the graphics help the readers better understand how the HyperSINDy framework works. There is also good discussion of the related works, i.e. ensembling methods and SDE-based approaches, which gives good motivation to the proposed method.
- **The proposed method is novel and achieves good improvements over existing methods.** It seems that the random differential equations (RDE) approach is a pretty novel perspective, and it is very natural to combine it with generative modeling. The HyperSINDy method also achieve uniformly better mean-squared error as well as uncertainty estimation than the best existing approach.

This paper seems like a solid advancement towards solving the very important problem of data-driven discovery of interpretable stochastic governing equations. This work will have wide applications in machine learning for science.

**Weaknesses:**

- **Experimental results on higher dimensional datasets might be a bit lacking.** One of the important claims of the advantage of HyperSINDy is that it circumvents the curse of dimensionality which hinders the performance of other methods. However, only the HyperSINDy results for one 10D system is given. It might be better if the authors can clarify how the other methods perform on this system, and/or give other examples of high dimensional systems.

**Questions:**

- In section 3, in "$H$ implements the implicit distribution $p_\theta(\mathbf{\Xi}|\mathbf{z})$", why is it the "implicit distribution?" From my understanding, shouldn't $\Xi_z$ just be a delta distribution (deterministic) on $H(z)$?
- $p_\theta(z)$ is modeled to be a standard Gaussian with diagonal covariance. Would the independence between different $z_t$ allow sudden jumps in the parameters of the system? Would it be better to model it as something like a Gaussian process?
- Related to the last question: does the discretization step size influence the model learning result?
- I don't think what "E-SINDy" stands for is ever introduced in the paper.

---

> ### Author Response · Authors · 2023-11-19
> **Rebuttal (Part 1/2)**
>
> ### Weaknesses
>
> > 1. Experimental results on higher dimensional datasets might be a bit lacking. One of the important claims of the advantage of HyperSINDy is that it circumvents the curse of dimensionality which hinders the performance of other methods. However, only the HyperSINDy results for one 10D system is given. It might be better if the authors can clarify how the other methods perform on this system, and/or give other examples of high dimensional systems.
>
> **Response:** Thank you for raising this point. Indeed, the ability to scale to higher dimensions is a distinguishing feature of HyperSINDy. To clarify, the limited scalability of existing SDE discovery methods precludes an assessment of their performance on the 10D system. However, we agree that it would still be informative to benchmark HyperSINDy’s performance against other methods for probabilistic model discovery (of deterministic ODEs) in this high-dimensional setting.
>
> To this end, we have updated the manuscript with new experiments applying E-SINDy to the 10D Lorenz system.
> In particular, we simulated the 10D Lorenz system with 3 different levels of noise on the forcing term ($\sigma=0$, $\sigma=5$, $\sigma=10$) and trained HyperSINDy and E-SINDy on each trajectory. We evaluated the RMSE of the mean and standard deviation of the discovered coefficients, as well as precision and recall. Refer to Table S5 in the appendix for the full results. Although E-SINDy achieves slightly improved performance in the noise-free case ($\sigma=0$), HyperSINDy outperforms E-SINDy across all metrics on  both of the higher noise levels.
>
> ### Questions
>
> > 1. In section 3, in "H implements the implicit distribution p(xi_z | z)", why is it the "implicit distribution?" From my understanding, shouldn't xi_z  just be a delta distribution (deterministic) on H(z)?
>
> **Response:* We apologize for the lack of clarity – we believe this reflects the distinction between z as a random sample vs. a random variable. For a specific sample $\hat{z}, Xi_{z} = H(z)$ is indeed deterministic. For the more general case corresponding to z as a random variable, p(Xi | z) refers to the posterior distribution of SINDy coefficients, conditioned on z. In this setting, the hypernetwork can be interpreted as an implicit distribution for p(Xi | z) (i.e. the neural network weights, biases, and architecture parameterize the distribution p(Xi | z), where z is a random variable). We use “implicit distribution” to mean that we do not actually have full access to the distribution of p(Xi | z), but can still sample from it. For example, if we sample an ensemble of z vectors, then feed that ensemble into H, we are effectively sampling from p(Xi | z). We have revised this line for clarity, including a pointer to (Pawlowski et al., 2018).

---

> > ### Comment · Reviewer_kqcu · 2023-11-21
> >
> > I thank the authors for their detailed responses. On the implicit distribution question, I think you confused the marginal distribution p(Xi) = \int p(Xi | z) p(z) dz that integrates out the zs (which can be approximated by sampling an ensemble of zs like you correctly pointed out), with the conditional distribution p(Xi | z) which is a delta distribution. It would be nice if the authors can fix this or provide further clarification on this point.

---

> > > ### Author Response · Authors · 2023-11-21
> > >
> > > Thank you very much for this clarification – we apologize for misinterpreting your earlier point. Indeed, p(Xi) is the marginal; p(Xi | z) is the conditional, which is a delta distribution. Thus, the hypernetwork serves as an implicit distribution for p(Xi). We have updated the Methods - Generative Model section to clarify this point.

---

> ### Author Response · Authors · 2023-11-19
> **Rebuttal (Part 2/2)**
>
> > 2. p(z) is modeled to be a standard Gaussian with diagonal covariance. Would the independence between different z_t allow sudden jumps in the parameters of the system? Would it be better to model it as something like a Gaussian process?
>
> **Response:** Yes, this independence could allow sudden jumps in the parameters of the system. However, because HyperSINDy learns to approximate the (potentially complex) distribution of states and derivatives (i.e., q_\phi(z|x,\dot{x} is kept close to the prior p_theta(z)), independent samples z_t are transformed into realizations of system coefficients that are similar to those observed in the original data. In other words, over the course of training, HyperSINDy learns to utilize i.i.d. dynamical noise to mimic the dynamical behavior of the observed data – including tendency toward sudden jumps. This is an important feature distinguishing HyperSINDy from other probabilistic methods that do not explicitly model the role of dynamical noise in the generation of the observed time series.
> We agree that Gaussian processes would provide an ideal framework for modeling the noise, particularly when moving beyond the i.i.d. noise case (see also (Casale et al., 2018; Mishra et al., 2022)). For our work, we assumed that the noise was i.i.d. in time, meaning that the independence of z_ts is a desirable modeling feature (as per the standard SDE framework). However, we note that extension to temporally correlated noise will be an important direction for future inquiry.
>
> > 3. Related to the last question: does the discretization step size influence the model learning result?
>
> **Response:** The discretization step size can influence results, although this is not an issue connected to HyperSINDy per se – rather, this relates to the more general and well-studied phenomenon of “finite-time effects” emerging in the numerical analysis of continuous stochastic dynamics based upon discretely sampled time series. Algorithmically correcting for such effects has been the topic of numerous works (e.g., (Ragwitz and Kantz, 2001; Lade, 2009; Honisch, 2011; Boujo and Cadot, 2019)), including one such approach recently introduced for stochastic SINDy (Callaham et al. 2021). Recent works also introduce promising approaches toward finite-time correction in the high-dimensional setting (e.g., (Brückner, Ronceray and Broedersz, 2020; Frishman and Ronceray, 2020)), suggesting a number of promising future directions for HyperSINDy to be extended to the theoretically continuous case.
>
> > 4. I don't think what "E-SINDy" stands for is ever introduced in the paper.
>
> **Response:** Thank you for pointing out this oversight! We have updated the Introduction to define E-SINDy.
>
> References
>
> Bauer, S. et al. (2017) ‘Efficient and Flexible Inference for Stochastic Systems’, in Advances in Neural Information Processing Systems. Curran Associates, Inc. Available at: https://papers.nips.cc/paper/2017/hash/e0126439e08ddfbdf4faa952dc910590-Abstract.html
>
> Boujo, E. and Cadot, O. (2019) ‘Stochastic modeling of a freely rotating disk facing a uniform flow’, Journal of Fluids and Structures, 86, pp. 34–43. Available at: https://doi.org/10.1016/j.jfluidstructs.2019.01.019.
>
> Brückner, D.B., Ronceray, P. and Broedersz, C.P. (2020) ‘Inferring the Dynamics of Underdamped Stochastic Systems’, Physical Review Letters, 125(5), p. 058103. Available at: https://doi.org/10.1103/PhysRevLett.125.058103.
>
> Casale, F.P. et al. (2018) ‘Gaussian Process Prior Variational Autoencoders’, in Advances in Neural Information Processing Systems.
>
> Frishman, A. and Ronceray, P. (2020) ‘Learning Force Fields from Stochastic Trajectories’, Physical Review X, 10(2), p. 021009. Available at: https://doi.org/10.1103/PhysRevX.10.021009.
>
> Honisch, C. (2011) ‘Estimation of Kramers-Moyal coefficients at low sampling rates’, Physical Review E, 83(6). Available at: https://doi.org/10.1103/PhysRevE.83.066701.
>
> Lade, S.J. (2009) ‘Finite sampling interval effects in Kramers–Moyal analysis’, Physics Letters A, 373(41), pp. 3705–3709. Available at: https://doi.org/10.1016/j.physleta.2009.08.029.
>
> Mishra, S. et al. (2022) ‘piVAE: a stochastic process prior for Bayesian deep learning with MCMC’, Statistics and Computing, 32(6), p. 96. Available at: https://doi.org/10.1007/s11222-022-10151-w.
>
> Pawlowski, N. et al. (2018) ‘Implicit Weight Uncertainty in Neural Networks’. arXiv. Available at: https://doi.org/10.48550/arXiv.1711.01297.
>
> Ragwitz, M. and Kantz, H. (2001) ‘Indispensable Finite Time Corrections for Fokker-Planck Equations from Time Series Data’, Physical Review Letters, 87(25), p. 254501. Available at: https://doi.org/10.1103/PhysRevLett.87.254501.
>
> Solin, A., Tamir, E. and Verma, P. (2021) ‘Scalable Inference in SDEs by Direct Matching of the Fokker– Planck– Kolmogorov Equation’, in Advances in Neural Information Processing Systems.

---

### Official Review · Reviewer_WJp7 · 2023-11-07

**Soundness:** 2 fair
**Presentation:** 3 good
**Contribution:** 2 fair
**Rating:** 5
**Confidence:** 5

**Summary:**

This work introduces HyperSINDy, a framework to model a family of special stochastic dynamics via a deep generative model of sparse, nonlinear governing equations whose parametric form is discovered from data.  HyperSINDy is built upon the combination of hypernetwork and SINDy and can learn a family of stochastic dynamics whose coefficients are driven by a Wiener process.

The main contributions of the HyperSINDy are summarized as follows:
(1) This framework can efficiently and accurately model random differential equations (random ODEs), whose coefficients are parameterized by a Wiener process. Hence, it provides a generative modeling of stochastic dynamics when their random ODE forms are driven by white noises.

(2) HyperSINDy can discover the analytical form of a sparse governing equation without a-priori knowledge. Also, by using the sparse masks, the computational complexity of HyperSINDy is scalable.

**Strengths:**

(1) The authors represent a proof of concept for this architecture, the manuscript is well written. The numerical results are convincing.
(2) The authors of this work employ the random differential equations (random ODEs) as the library of candidate functions for SINDy. This approach is innovative and it enables the extension of SINDy from deterministic to stochastic dynamics.

**Weaknesses:**

(1) Random differential equations are conjugate to stochastic differential equations. It is unclear how to convert a general SDEs into its random ODEs representations, for example, the Langevin type dynamics.
(2) This manuscript lacks a comparison with other methods.
(3) Although the authors have commented in the manuscript, it is still unclear if this HyperSINDy framework can handle complex noise terms as well as the robustness of noises.

**Questions:**

(1) This manuscript could have been enhanced if it can provide examples of learning underdamped Langevin systems, for example, learn the harmonic oscillator under  thermal bath.
(2) In particular, the manuscript could have been enhanced if it can provide an appendix discussion on how to construct a Random ODE representation for a general SDE.
(3) The manuscript could have been enhanced if it can provide numerical examples when different types of noises are added to the observation data.

---

> ### Author Response · Authors · 2023-11-19
>
> ### Weaknesses
> > 1. Random differential equations are conjugate to stochastic differential equations. It is unclear how to convert a general SDEs into its random ODEs representations, for example, the Langevin type dynamics.
>
> **Response:** Although this conjugacy is well-established, explicit transformations between SDEs and RDEs are not always straightforward. The standard procedure involves substitution via an Ornstein-Uhlenbeck process, the solution to a linear SDE. Following Han & Kloeden (2017), we may demonstrate this transformation for a scalar SDE with additive noise:
>
> $$
> \begin{align}
> 	dX_t = f(X_t)dt + dWt
> \end{align} $$
> becomes
> $$\begin{align}
> 	\dot{Z_t} = f(Z_t + O_t) + O_t,
> \end{align} $$
> where $Z_t \coloneqq X_t - O_t$ and $O_t$ is the stationary Ornstein-Uhlenbeck process satisfying the SDE $dO_t = -O_tdt + dW_t$.
>
> A similar conjugation can work for many cases of additive or multiplicative SDEs. In general, though, this conversion is not straightforward to implement. Thus, although we appeal to RDE-SDE conjugacy in order to validate results obtained in the RDE framework, the ability to convert between these representations in practice is more challenging.
>
> On the other hand, this challenge of converting between SDE and RDE formulations highlights unique advantages of our problem formulation. Indeed, much of the interest in RDEs is motivated from the practical advantages afforded by this formulation in certain scenarios in comparison to SDEs (e.g., (Bauer et al., 2017; Solin, Tamir and Verma, 2021)).
>
> We have added this discussion to the Supplementary Appendix.
>
> > 2. This manuscript lacks a comparison with other methods.
>
> **Response:** Our original submission included comparisons with the leading “ensemble SINDy” method for robust equation discovery in the presence of noise, and with “stochastic SINDy” for a two-dimensional SDE discovery model. We have updated the manuscript to also include comparisons with Bayesian Spline Learning. Refer to Table S6 in the appendix for comparisons between HyperSINDy, E-SINDy, and Bayesian Spline Learning on a Lotka-Volterra RDE simulation.
>
> > 3. Although the authors have commented in the manuscript, it is still unclear if this HyperSINDy framework can handle complex noise terms as well as the robustness of noises.
>
> **Response:** We have performed new experiments to address the question of how our method performs under different scenarios for the noise. Refer to Figure S5 in the appendix for results comparing HyperSINDy and E-SINDy on a Lotka-Volterra system simulated as an RDE, but where each coefficient is drawn from a Half-Normal distribution. In this case, the distribution of coefficients does not match the Gaussian assumption of the latent space; however, HyperSINDy is still able model the complex coefficient distributions due to the expressivity provided by its neural network. This result establishes that the method has potential to generalize beyond the iid noise case, while leaving for subsequent work a more systematic validation in different noise settings.
>
> ### Questions
> > 1. This manuscript could have been enhanced if it can provide examples of learning underdamped Langevin systems, for example, learn the harmonic oscillator under thermal bath.
>
> **Response:** Thank you for raising these possibilities for further extensions of our approach. Our updated Appendix clarifies that explicit RDE-SDE transformation is not always trivial, meaning the recasting of these canonical physics SDE problems may not be entirely straightforward; however, we agree that it represents an important future direction for broadening the scope of our framework.
>
> > 2.  In particular, the manuscript could have been enhanced if it can provide an appendix discussion on how to construct a Random ODE representation for a general SDE.
>
> **Response:** Thank you for making this suggestion, which we agree aids the interpretation of our framework. We have updated the appendix as requested (see also response to Weaknesses #1 above).
>
> > 3. The manuscript could have been enhanced if it can provide numerical examples when different types of noises are added to the observation data.
>
> **Response:** Please see response to Weaknesses #3 above.
>
> References
>
> Bauer, S. et al. (2017) ‘Efficient and Flexible Inference for Stochastic Systems’, in Advances in Neural Information Processing Systems. Curran Associates, Inc. Available at: https://papers.nips.cc/paper/2017/hash/e0126439e08ddfbdf4faa952dc910590-Abstract.html
>
> Han, X. and Kloeden, P.E. (2017) Random Ordinary Differential Equations and Their Numerical Solution. Singapore: Springer Singapore (Probability Theory and Stochastic Modelling). Available at: https://doi.org/10.1007/978-981-10-6265-0.
>
> Imkeller, P. and Schmalfuss, B. (2001) ‘The Conjugacy of Stochastic and Random Differential Equations and the Existence of Global Attractors’, Journal of Dynamics and Differential Equations, 13(2), pp. 215–249.

---

### Official Review · Reviewer_7UM6 · 2023-11-08

**Soundness:** 2 fair
**Presentation:** 3 good
**Contribution:** 2 fair
**Rating:** 5
**Confidence:** 3

**Summary:**

Authors introduce a framework called HyperSINDy for modeling stochastic dynamics using a deep generative model that discovers the parametric form of sparse governing equations from data. It employs an inference model and generative model to discover
an analytical representation of observed stochastic dynamics in the form of a random ODE (RODE). It is particularly useful for random coefficients.

**Strengths:**

Figure 1 shows the scheme of this method. It has three steps: inference mode, generative model and SINDy. It basically glue the Hypernetwork and SINDy together to tackle the random coefficient case.

**Weaknesses:**

1. It is a typical A+B type of paper. Each part is well studied and author glue them together and demonstrate it in several simple examples. I don't think there is enough novelty here.

2. All three examples are artificially made for this algorithm. All examples are corrected identified but I am not impressed unless authors are able to demonstrate some non-trivial RODE. The second example equation (11) is not even a valid example of stochastic  Lotka-Volterra. I don't know what is N(0,1) on the Right hand side means here.

3. Authors have limited knowledge on RODE here in fact not all SDE can be transformed to RODE and vice versa. And in general RODE case, z is not independent with x.

**Questions:**

If x' is not available (e.g., after training), z is sampled from the prior z ∼ p_θ(z) to produce \Xi. I don't understand this part. Please elaborate more or give an example.

---

> ### Author Response · Authors · 2023-11-19
> **Rebuttal (Part 1/2)**
>
> Thank you for your comments on how the manuscript can be made more compelling. We wish to suggest that there are a couple points of confusion here – we apologize for the lack of clarity and have done our best to address these points in the revised manuscript.
>
> ### Weaknesses
> > 1. It is a typical A+B type of paper. Each part is well studied and author glue them together and demonstrate it in several simple examples. I don't think there is enough novelty here.
>
> **Response:** We believe the present contribution goes well beyond a “typical A+B paper”, in the sense that the two methods claimed to be simply glued together (hypernetworks (Ha, Dai and Le, 2016; Pawlowski et al., 2018) and SINDy (Brunton, Proctor and Kutz, 2016; Kaptanoglu et al., 2022) are just two pieces of an overarching conceptual and technical framework being put forward. Moreover, these two methods refer to high-level frameworks/architectures with many possible implementations (e.g., a hypernetwork is simply a neural network that predicts the weights of another network – a concept we reimagine for predicting the coefficients of an analytical expression). Thus, the integration of the two methods (which have arisen in different academic communities for very different purposes) is far from straightforward.
>
> At a conceptual level, integration of these methods is entirely contingent on our innovative problem formulation in terms of random differential equations. This alternative formulation for stochastic dynamics enabled us to leverage the unique advantages of numerous algorithms/techniques (including SINDy and hypernetworks, but also, e.g., variational autoencoders and backpropagation-compatible L0 loss) toward our overarching goal of stochastic dynamical modeling.
>
> At a technical level, unifying all these approaches within a common framework is a highly non-trivial task, requiring strategies to appropriately balance multiple losses relating to equation sparsity, overall model complexity, reconstruction error, and proximity to a Gaussian prior (that is, promoting discovery of a model approximating a canonical SDE).
>
> All in all, while it can be heuristically useful to summarize the proposed approach as combining hypernetworks and SINDy, this intuitive picture belies a much more sophisticated integration of techniques and ideas. We believe that this claim is consistent with evaluations from the other reviews, who each note the novelty/innovation of the proposed framework.
>
> > 2. All three examples are artificially made for this algorithm. All examples are corrected identified but I am not impressed unless authors are able to demonstrate some non-trivial RODE. The second example equation (11) is not even a valid example of stochastic Lotka-Volterra. I don't know what is N(0,1) on the Right hand side means here.
>
> **Response:** We believe that this impression stems in part from notation differences between fields. We have modeled systems subject to white noise (thus, a general setting for stochastic dynamics), though we adopt notation more familiar in statistics/machine learning. Thus, N(0,1) denotes a Gaussian distribution with mean=0, standard deviation=1; iteratively sampling from this distribution amounts to simulating a specific realization of a driving white noise process (which is defined by random variables distributed according to N(0,1) at each time point t (Han and Kloeden, 2017). As such, our selected problems are consistent with widely used stochastic modeling problems (namely, classic SDEs) in which the source of stochasticity is taken to be a Gaussian white noise. Nonetheless, we agree that this notation can obscure connections to standard SDE representations; as such, we have rewritten equation 11 in canonical SDE matrix notation involving a 2D Wiener process (which we simulate via the Euler-Maruyama integration scheme), thus clarifying that it is indeed a valid Lotka-Volterra SDE.
>
> > 3. Authors have limited knowledge on RODE here in fact not all SDE can be transformed to RODE and vice versa. And in general RODE case, z is not independent with x.
>
> **Response:** As we understand, any finite-dimensional, ordinary SDE can be transformed to an RDE; we agree that the reciprocal is more restricted to RDEs driven by stochastic processes that are the solutions of SDEs (Han and Kloeden, 2017). To our knowledge, this reciprocity is well-established since the results of (Imkeller and Schmalfuss, 2001), and has been usefully invoked in other numerical investigations into stochastic dynamics where an RDE formulation carries practical advantages (e.g., (Bauer et al., 2017)).
> We agree that in the general case, the noise process z(t) need not be state-independent. However, the state-independent case (i.e., an autonomous noise process) represents the most common problem setting for stochastic dynamics, and thus carries broad relevance beyond the specific models studied herein.

---

> > ### Author Response · Authors · 2023-11-19
> > **Rebuttal (Part 2/2)**
> >
> > ### Questions
> >
> > > 1. If x' is not available (e.g., after training), z is sampled from the prior z ∼ p_θ(z) to produce \Xi. I don't understand this part. Please elaborate more or give an example.
> >
> > **Response:** This statement refers to the (post-training) use of HyperSINDy to generate stochastic dynamics. Thus, new trajectories are generated by simply sampling z from the prior p_θ(z) (a standard gaussian with diagonal covariance, in the present setting), then feed this z into the hypernetwork to produce a prediction of \Xi. We appreciate you bringing this potentially unclear text to our attention; we have updated this text (Fig. 1 caption) to clarify the above.
> >
> > References
> >
> > Bauer, S. et al. (2017) ‘Efficient and Flexible Inference for Stochastic Systems’, in Advances in Neural Information Processing Systems. Curran Associates, Inc. Available at: https://papers.nips.cc/paper/2017/hash/e0126439e08ddfbdf4faa952dc910590-Abstract.html
> >
> > Brunton, S.L., Proctor, J.L. and Kutz, J.N. (2016) ‘Discovering governing equations from data by sparse identification of nonlinear dynamical systems’, Proceedings of the National Academy of Sciences, 113(15), pp. 3932–3937. Available at: https://doi.org/10.1073/pnas.1517384113.
> >
> > Ha, D., Dai, A. and Le, Q.V. (2016) ‘HyperNetworks’. arXiv. Available at: http://arxiv.org/abs/1609.09106 (Accessed: 9 July 2022).
> >
> > Han, X. and Kloeden, P.E. (2017) Random Ordinary Differential Equations and Their Numerical Solution. Singapore: Springer Singapore (Probability Theory and Stochastic Modelling). Available at: https://doi.org/10.1007/978-981-10-6265-0.
> >
> > Imkeller, P. and Schmalfuss, B. (2001) ‘The Conjugacy of Stochastic and Random Differential Equations and the Existence of Global Attractors’, Journal of Dynamics and Differential Equations, 13(2), pp. 215–249. Available at: https://doi.org/10.1023/A:1016673307045.
> >
> > Kaptanoglu, A.A. et al. (2022) ‘PySINDy: A comprehensive Python package for robust sparse system identification’, Journal of Open Source Software, 7(69), p. 3994. Available at: https://doi.org/10.21105/joss.03994.
> >
> > Pawlowski, N. et al. (2018) ‘Implicit Weight Uncertainty in Neural Networks’. arXiv. Available at: https://doi.org/10.48550/arXiv.1711.01297.

---

> > ### Comment · Reviewer_7UM6 · 2023-11-20
> > **Unclear how to turn stochastic Lotka-Volterra system into RODE**
> >
> > Authors have changed the model for the second experiment. However, in appendix F, they only mentioned how to transform the additive noise version. But the stochastic Lotka-Volterra has multiplicative noise. Does the OU transform still work here?
> > Do author claim their methods can work for arbitrary SDE?

---

> > > ### Author Response · Authors · 2023-11-20
> > >
> > > Thank you for your prompt follow-up. We wish to clarify that we did not change the original second experiment, which already involved multiplicative (state-dependent) noise – we simply re-wrote in canonical SDE form.
> > >
> > > Transformation is indeed possible for multiplicative noise. Thus, following the presentation from Han & Kloeden (2017) (Section 3.5.1), the SDE with multiplicative noise:
> > >
> > > $$
> > > \begin{align}
> > > 	dX_t = f(t, X_t)dt + b(t)X_tdWt
> > > \end{align}
> > > $$
> > >
> > > may be combined with the random transformation
> > >
> > > $$
> > > \begin{gather}
> > > z(t) = T(t)X_t, \\
> > > T(t) := \exp{ \left( \frac{1}{2} \int_{0}^{t} b^2(s) \,ds - \int_{0}^{t} b(s) \,dW_s \right)}
> > > \end{gather}
> > > $$
> > >
> > > to obtain the RDE:
> > >
> > > $$
> > > \begin{align}
> > > \frac{dz}{dt} = T(t)f\left(t,T^{-1}(t)z(t) \right).
> > > \end{align}
> > > $$
> > >
> > > (see also (Imkeller and Schmalfuss, 2001; Neckel and Rupp, 2013; Caraballo and Han, 2016) for further discussion of this topic).
> > >
> > > That said, we wish to stress that our manuscript does not claim the ability to explicitly map between particular RDE and SDE expressions, and our results are not contingent on the ability to carry out this transformation. Rather, in general,  the chief motivation for stochastic equations (whether SDEs or RDEs) lies simply in the mathematical modeling of dynamics under uncertainty; for many (if not most) modeling applications, the precise physical nature of this uncertainty – e.g., a diffusion process or fluctuating system parameters – is unknown and/or of secondary importance (Friedrich et al., 2011; Duan, 2015; Särkkä and Solin, 2019). In this sense, SDEs and RDEs simply emerge as alternative modeling frameworks, each with unique practical (dis)advantages that must be considered in context. For our purposes, we simply note that the conjugacy between the two formulations means that the results obtained in one framework are recognized to be directly pertinent to those obtained in the other (Han and Kloeden, 2017).
> > >
> > > We will update the Appendix to include this additional transformation and clarifying comments.
> > >
> > > References
> > >
> > > Caraballo, T. and Han, X. (2016) Applied Nonautonomous and Random Dynamical Systems. Cham: Springer International Publishing (SpringerBriefs in Mathematics). Available at: https://doi.org/10.1007/978-3-319-49247-6.
> > >
> > > Duan, J. (2015) An Introduction to Stochastic Dynamics. 1st edition. New York, NY: Cambridge University Press.
> > >
> > > Friedrich, R. et al. (2011) ‘Approaching complexity by stochastic methods: From biological systems to turbulence’, Physics Reports, 506(5), pp. 87–162. Available at: https://doi.org/10.1016/j.physrep.2011.05.003.
> > >
> > > Han, X. and Kloeden, P.E. (2017) Random Ordinary Differential Equations and Their Numerical Solution. Singapore: Springer Singapore (Probability Theory and Stochastic Modelling). Available at: https://doi.org/10.1007/978-981-10-6265-0.
> > >
> > > Imkeller, P. and Schmalfuss, B. (2001) ‘The Conjugacy of Stochastic and Random Differential Equations and the Existence of Global Attractors’, Journal of Dynamics and Differential Equations, 13(2), pp. 215–249. Available at: https://doi.org/10.1023/A:1016673307045.
> > >
> > > Neckel, T. and Rupp, F. (2013) ‘Random Differential Equations in Scientific Computing’, in Random Differential Equations in Scientific Computing. De Gruyter Open Poland. Available at: https://doi.org/10.2478/9788376560267.
> > >
> > > Särkkä, S. and Solin, A. (2019) Applied Stochastic Differential Equations. 1st edn. Cambridge University Press. Available at: https://doi.org/10.1017/9781108186735.

---

### Author Response · Authors · 2023-11-19
**Overall response**

We sincerely thank the reviewers for their valuable feedback, which we have used to significantly improve the manuscript. Please consult the current (revised) version of the manuscript, which incorporates the suggested clarifications and new experiments.

Most substantially, we have performed numerous additional experiments to address the concerns/requests raised by reviewers. These experiments address requests for additional quantitative metrics, additional comparisons against state-of-the-art methods, extension to non-Gaussian noise, and evaluation of the dependence of our method’s performance on the selected latent dimension. Details of these experiments are contained in Appendix H. The new figures and tables we have introduced in the appendix are as follows:

-- Table S4, which reports precision and recall metrics for HyperSINDy and E-SINDy models. These tables further quantitatively support the model discovery capabilities of HyperSINDy. \
-- Table S5, which compares HyperSINDy and E-SINDy on the 10D Lorenz-96 system. The table reports RMSE, precision, and recall results, further establishing HyperSINDy’s capacity for equation discovery in the high-dimensional setting. \
-- Figure S5: Simulating of a Lotka-Volterra RDE, but with coefficients drawn from a HalfNormal (rather than Gaussian) distribution. This figure demonstrates the feasibility of HyperSINDy for modeling more complex (non-Gaussian) noise, highlighting the power of the neural network (i.e., the hypernetwork) for learning potentially complex coefficient distributions, as compared to E-SINDy. \
-- Table S6, which compares performance of HyperSINDy, E-SINDy, and Bayesian Spline Learning (Sun et al., 2022) on a Lotka-Volterra RDE across a range of (i.i.d.) noise levels. We train HyperSINDy, E-SINDy, and Bayesian Spline Learning on these trajectories. Table S6 shows RMSE, precision, and recall results for these models. These results establish favorable performance of HyperSINDy against an additional state-of-the-art method for Bayesian equation discovery. \
-- Figure S6, which evaluates HyperSINDy performance on the stochastic Lorenz system as a function of z dimension. This figure establishes that HyperSINDy’s favorable performance is not highly sensitive to the specified latent dimension. \

In addition to these new experiments, we have edited the manuscript text for clarity; we believe these clarifications address a number of additional points raised by reviewers, altogether resulting in a significantly improved manuscript.

Below, we respond in detail to each of the questions and concerns raised by reviewers. We look forward to any further discussion of these points.

---

### Author Response · Authors · 2023-11-21

Dear Reviewers,

We have updated the Appendix to include the latest points arising in this discussion period.

With the review window coming to a close, we wish to thank you again for all of your helpful feedback on the manuscript, which has been critical in helping us further improve the quality of our paper. We believe our numerous added experiments, metrics, and improved clarity have fully addressed the main concerns about the manuscript. We hope to receive any additional feedback, and we would greatly appreciate your potential re-consideration of original ratings in light of our revisions and clarifications.

Sincerely,
Authors

---

### Meta-Review · Area_Chair_WfHm · 2023-12-05

**Metareview:**

SINDy is a method that learns deterministic differential equation from time series data based on dictionary learning and LASSO, and this paper proposes a way to extend it to stochastic settings. VAE and L0 regularization techniques are combined with SINDy to enable this extension. It is reviewed by expert reviewers. While some reviewers considered the paper to have exhibited novelty in both technical and conceptual angles, several other reviewers expressed concerns about novelty and applicability. Unfortunately the disagreement did not get resolved till the end of the reviewer-AC discussion period. Based on carefully checking each reviewer's assessment, it seems a more convincing, concrete description on how to construct a Random ODE representation for a general SDE is still missing, which however is an essential premise of the work. Therefore, I'm not sure if I can recommend acceptance at the current stage. However, I hope the authors could find the reviews helpful and consider a resubmission after revision.

**Justification For Why Not Higher Score:**

Multiple expert reviewers independently raised similar concerns about novelty and applicability.

**Justification For Why Not Lower Score:**

N/A

---

### Decision · Program_Chairs · 2024-01-16

Reject